# Continuous artificial synthesis of glucose precursor using enzyme-immobilized microfluidic reactors

Yujiao Zhu [1,2], Ziyu Huang [3], Qingming Chen [1], Qian Wu[4,5], Xiaowen Huang [3], Pui-Kin So[6], Liyang Shao [7], Zhongping Yao [4,5], Yanwei Jia [2,8,9], Zhaohui Li[10,11], Weixing Yu [12], Yi Yang [13], Aoqun Jian [14], Shengbo Sang [14], Wendong Zhang [14] & Xuming Zhang [1]

Food production in green crops is severely limited by low activity and poor specificity of D-ribulose-1,5-bisphosphate carboxylase/oxygenase (RuBisCO) in natural photosynthesis (NPS). This work presents a scientific solution to overcome this problem by immobilizing RuBisCO into a microfluidic reactor, which demonstrates a continuous production of glucose precursor at 13.8 $\mu$mol g$^{-1}$ RuBisCO min$^{-1}$ from $CO_2$ and ribulose-1,5-bisphosphate. Experiments show that the RuBisCO immobilization significantly enhances enzyme stabilities (7.2 folds in storage stability, 6.7 folds in thermal stability), and also improves the reusability (90.4% activity retained after 5 cycles of reuse and 78.5% after 10 cycles). This work mimics the NPS pathway with scalable microreactors for continuous synthesis of glucose precursor using very small amount of RuBisCO. Although still far from industrial production, this work demonstrates artificial synthesis of basic food materials by replicating the light-independent reactions of NPS, which may hold the key to food crisis relief and future space colonization.

[1] Department of Applied Physics, The Hong Kong Polytechnic University, Hong Kong, China. [2] State Key Laboratory of Analog and Mixed Signal VLSI, Institute of Microelectronics, University of Macau, Macau, China. [3] State Key Laboratory of Biobased Material and Green Papermaking; Shandong Provincial Key Laboratory of Microbial Engineering, Department of Bioengineering, Qilu University of Technology (Shandong Academy of Sciences), Jinan, China. [4] State Key Laboratory for Chemical Biology and Drug Discovery, Food Safety and Technology Research Centre, Department of Applied Biology and Chemical Technology, The Hong Kong Polytechnic University, Hong Kong, China. [5] State Key Laboratory of Chinese Medicine and Molecular Pharmacology (Incubation), Shenzhen Key Laboratory of Food Biological Safety Control, Shenzhen Research Institute of the Hong Kong Polytechnic University, Shenzhen, China. [6] University of Research Facility in Life Sciences, The Hong Kong Polytechnic University, Hong Kong, China. [7] Department of Electrical and Electronic Engineering, Southern University of Science and Technology, Shenzhen, China. [8] Faculty of Science and Technology, University of Macau, Macau, China. [9] Faculty of Health Sciences, University of Macau, Macau, China. [10] School of Electronics and Information Engineering, Sun Yat-Sen University, Guangzhou, China. [11] Southern Marine Science and Engineering Guangdong Laboratory (Zhuhai), Zhuhai, China. [12] Key Laboratory of Spectral Imaging Technology, Xi'an Institute of Optics and Precision Mechanics, Chinese Academy of Sciences, Xi'an, China. [13] School of Physics & Technology, Wuhan University, Wuhan, China. [14] MircoNano System Research Center, College of Information and Computer Science, Taiyuan University of Technology, Taiyuan, China. Correspondence and requests for materials should be addressed to X.Z. (email: apzhang@polyu.edu.hk)

The world is moving rapidly into food crisis due to extreme weather, shortage of farmland, and population explosion. And glucose is the basic material of food produced by green plants using the natural photosynthesis (NPS)[1]. However, the NPS suffers from low-energy efficiency (~1%) and is greatly limited by soil, climate, water, and labor[2,3]. It is reported that food demand is expected to increase by 70% by 2050[4], which makes an extremely urgent demand for a scientific solution beyond NPS for food production. Recently, scientists have figured out an excellent alternative to NPS, which is called artificial photosynthesis (APS), for the energy-rich chemicals production. It mimics the green plants to produce energy-rich materials and fuels but using man-made nanomaterials and engineered photosynthetic reactors. Compared with the NPS, the APS usually presents much higher efficiency (~12%) and simplicity[5]. Although the APS has attracted substantial research interest, most of efforts have been devoted to the light-dependent reactions for hydrogen generation and water splitting[6,7], microbial growth and biofuel production[8–12], and photocatalytic cofactor regeneration[13–16]. The exploration of light-independent reactions for food production is worth more attentions. Nevertheless, it is found that the light-independent reactions, though crucial for glucose production from $CO_2$, are more challenging for the artificial replication, since they involve multi-step enzymatic reactions (Fig. 1a).

The first carbon fixation phase in light-independent reactions is catalyzed by the most abundant enzyme in plants, RuBisCO (EC 4.1.1.39). Even so, the low catalytic rate and the poor specificity of RuBisCO seriously limit the agricultural productivity[17]. Although many studies have been conducted to improve the efficiency of RuBisCO[18–23], they are mainly designed for in vivo biological systems[24,25]. Lots of energy and labor are consumed to balance the metabolic pathways in intact cells. For the industrial cell-free application with the demands of sustainability and scalability, immobilizing and concentrating RuBisCO into man-made structures presents to be a more promising method to enhance the overall catalytic efficiency[26–28]. Enzyme immobilization has many advantages, such as high enzyme-to-reactant ratio, enhanced enzyme stability, admirable enzyme reusability, weakened feedback inhibition, and possible modulation of catalytic property[29–31]. Various enzyme immobilization techniques have been thoroughly explored, such as physical adsorption, affinity bonding, covalent binding, and encapsulation. However, the choice of techniques should be very careful, since inappropriate immobilization may cause the reduction of enzyme activity due to conformational change, detachment, and inactivation[32]. Among all the enzyme immobilization methods, covalent immobilization always offers the strongest bond between enzyme and support[33,34]. But it usually involves complex linkers and organic solvents. Recently, polydopamine (PDA) becomes rather popular in biomimetic systems for the covalent immobilization of enzymes thanks to their excellent biocompatibility and high efficiency in imparting biological functionality on inorganic materials[35–38]. Moreover, the PDA functionalized on smooth substrates would form a coarse surface assembled by many particles, which could enlarge the active surface for immobilization[37,38].

In plants, the NPS occurs in a natural microfluidic system of microscale chloroplasts and fluids-filled areas[39]. In laboratory research, scientists usually make use of microfluidic systems to mimic the reactions and behaviors in natural instead of bulk systems. Thereby, the inherent advantages of microfluidics can be well taken, such as low sample consumption, flexible manipulation, large surface-area-to-volume ratio, and self-refreshing surface effect[40]. In addition, remarkable enhancement of the overall reaction production can also be expected[41–43]. Moreover, when the enzyme is immobilized into microfluidic reactors, separation and recovery of enzyme can be achieved simultaneously along with the product generation, avoiding the contamination and reducing the enzyme consumption[44,45].

On account of the above benefits, this work plans to constitute the NPS system for glucose precursor (3-PGA) production by replicating the first phase reaction of NPS using microfluidic reactors. PDA is employed as the linker to immobilize RuBisCO on the inner surfaces of reactors. The reaction performance of artificial chemosynthesis of glucose precursor in the RuBisCO-immobilized microfluidic reactors (RIMRs) is compared with that in the bulk reaction in terms of the stability, reusability, and long-term 3-PGA production subsequently. This is a proof of concept for constructing the NPS pathway using enzyme-immobilized microfluidic reactors for continuous synthesis of glucose precursor.

## Results

**Fabrication and characterization of RIMRs**. The three-dimensional diagram and the photograph (inset) of the RIMRs is presented in Fig. 1b. It consists of two PDMS layers, and the inner surfaces of microchannels are functionalized by PDA. The abundant catechol groups of PDA layer can react with the amine groups of RuBisCO through the Michael addition or Schiff base reactions[35], facilitating the immobilization of RuBisCO (see Supplementary Fig. 1). Figure 1c is the SEM image of the microchannels' inner surfaces after the RuBisCO immobilization. The PDA-modified PDMS is coarse and rough and is covered with many PDA nanoparticles. The RuBisCO immobilization then leads to the formation of some blocks, proving the successful immobilization (see Supplementary Fig. 3). The successful immobilization of RuBisCO is further confirmed by the measurements of water contact angle, Raman and ATR-FTIR spectrum and fluorescence images of the microfluidic channels (see Supplementary Figs. 4–6).

**Protein-loading capacity and kinetic parameters**. After the successful immobilization of RuBisCO, the protein-loading capacity of the PDA-modified microfluidic reactors is investigated to determine the optimal RuBisCO concentration for further experiments. As plotted in Fig. 1d (dark solid squares), the amount of protein loaded in the reactors rises with the increase of the injected RuBisCO concentration. It then saturates to about $2.0\,\mu g\,cm^{-2}$ when the RuBisCO concentration reaches about $6.25\,\mu g\,\mu L^{-1}$. However, the protein-loading efficiency decreases with the increase of RuBisCO concentration (red open triangles in Fig. 1d). This shows that RuBisCO can be easily immobilized onto the PDA layer. Once all the functional groups of the PDA are covalently bonded with RuBisCO, the excessive RuBisCO would be easily rinsed off from the reactor. Consequently, $6.25\,\mu g\,\mu L^{-1}$ is chosen as the optimal RuBisCO concentration for the following RIMRs experiments. The protein-loading capacity of the PDA-modified microfluidic reactor is estimated to be $2.03\,\mu g\,cm^{-2}$, and the protein-loading efficiency is ~50%. In the following experiments, the RuBisCO amounts used for both the immobilized and the free RuBisCO test are $21.875\,\mu g$.

Then, the kinetic parameters of the immobilized RuBisCO in the RIMRs reaction and the free RuBisCO in the bulk reaction are determined (see Table 1). Compared with the kinetic parameters of free RuBisCO, the $K_m$ value (the Michaelis–Menten constant) for the RIMRs is relatively higher (0.070 vs. 0.049 mM), inferring a lower affinity of the immobilized enzyme for the reactant. This is a common and intrinsic weakness of immobilized enzymes, owing to the steric hindrance introduced by the coverage of some active site by the support[46]. In addition, the immobilized enzyme

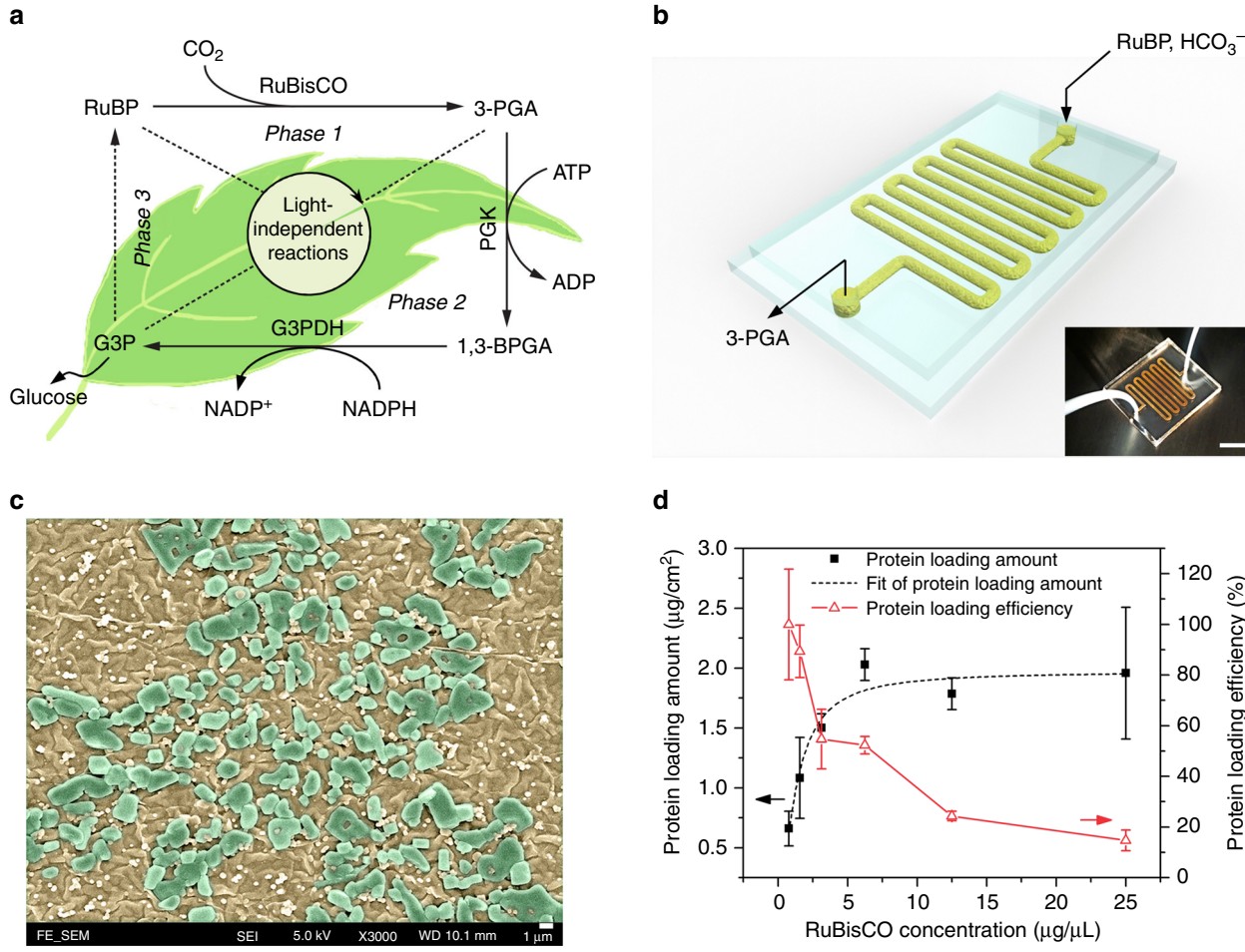

**Fig. 1** Design and characterization of RuBisCO immobilized microfluidic reactors (RIMRs). **a** Scheme of light-independent reactions of NPS: *Phase 1*: Carbon fixation starts with ribulose 1,5-bisphosphate (RuBP) and uses the enzyme RuBisCO to fix $CO_2$ into 3-phosphoglycerate (3-PGA); *Phase 2*: Reduction reaction uses adenosine triphosphate (ATP), nicotinamide adenine dinucleotide phosphate (NADPH) and the enzyme phosphoglycerate kinase (PGK) and glyceraldehyde 3-phosphate dehydrogenase (G3PDH) to reduce 3-PGA into glyceraldehyde 3-phosphate (G3P), two of which can form the end product glucose; *Phase 3*: RuBP regeneration from G3P using up to 9 steps of enzymatic reactions. **b** Three-dimensional diagram and the photograph (inset) of the RIMRs, the scale bar of the inset is 1 cm. **c** SEM image of the inner surfaces of RIMRs. Flat and smooth PDMS becomes rough and is covered by PDA nanoparticles after PDA modification (brownish color). Large blocks are the immobilized RuBisCO (green blocks). The scale bar is 1 μm. **d** Protein-loading amount and protein-loading efficiency as a function of the concentrations of injected RuBisCO to find the optimal RuBisCO concentration for further experiments. Error bars represent the standard deviations from three independent experiments. Source data are provided as a Source Data file

### Table 1 Kinetic parameters obtained from RIMRs reaction and bulk reaction[a]

| Reaction type | $V_{max}$ (mmol min$^{-1}$ g$^{-1}$ RuBisCO) | $K_m$ (RuBP) (mM) |
|---|---|---|
| RIMRs reaction | 0.070 ± 0.003 | 0.070 ± 0.012 |
| Bulk reaction | 0.169 ± 0.006 | 0.049 ± 0.008 |

[a]The collected production solutions I and II are both 100 μL. RuBP concentrations are 0.01–2 mM for the bulk reaction and 0.025–2 mM for the RIMRs reaction. The concentration of bicarbonate ($HCO_3^-$) in the reaction buffer is 66 mM. $K_m$ and $V_{max}$ values are the means ± s.d. of three independent experiments, which is calculated by the GraphPad Prism 7 according to the nonlinear fitting of Michaelis–Menten model. Source data are provided as a Source Data file

may lose the flexibility to bind the natural ligands during catalysis and the diffusion distance for the reactant to the immobilized enzyme on the microchannel sidewalls becomes relatively long (as compared to the free enzyme)[46]. From Table 1, the RIMRs also exhibited a lower $V_{max}$ (the maximal reaction rate) than that of the free RuBisCO. The activity loss induced by the immobilization procedure is ascribed to the conformational changes of

RuBisCO. Such conformational changes can also be observed from the presence of a weak band at about 1600 cm$^{-1}$ of RuBisCO-PDA-PDMS in Supplementary Fig. 5b. According to the previous work, the mechanism of PDA for the enzyme immobilization is to link the amino groups of enzymes to the catechol groups of PDA[35,38]. It has been reported that the amino groups of lysine residues at the active sites of RuBisCO play a critical role in catalysis[47,48]. Therefore, the immobilization of RuBisCO by PDA modification may cause irreversible conformational changes of RuBisCO and thus result in the reduction of its activity.

It is noted that the activity of free RuBisCO is very low compared with the reported RuBisCO activity before. According to Table 1, the $k_{cat}$ (turnover number) of free RuBisCO here is about 0.19 s$^{-1}$, which is remarkably smaller than the natural $k_{cat}$ of RuBisCO from Spinach (~3 s$^{-1}$)[49]. One reason is likely the irreversible binding of inhibitory contaminants produced during the commercial preparation of RuBP[50,51]. Another reason for the low activity of RuBisCO here may be the fallover during catalysis, which is caused by the tight binding of inhibitory sugar phosphates to the active site[52–54].

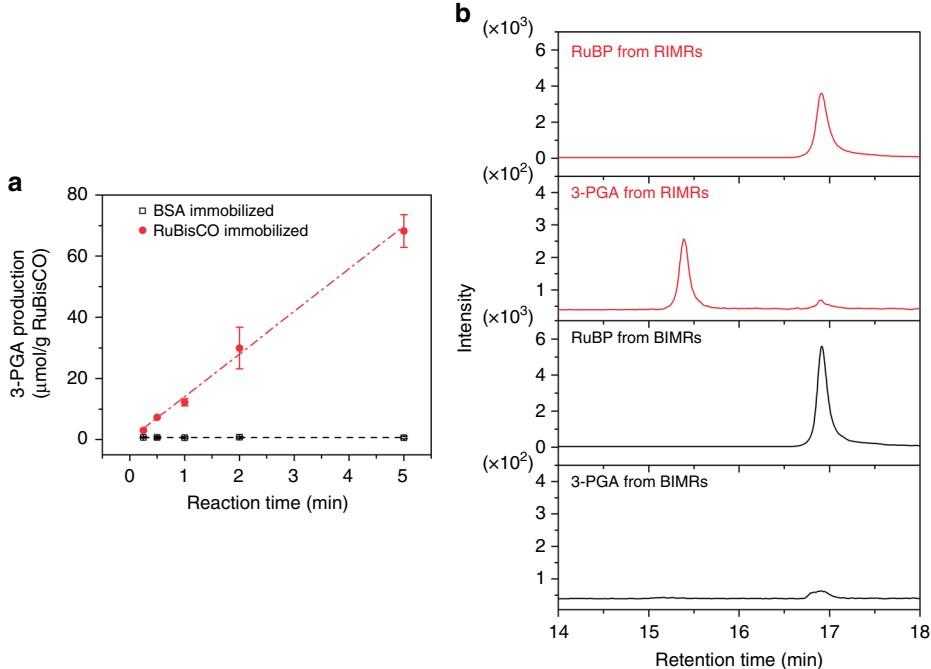

**Fig. 2** Feasibility of RIMRs. **a** Production amount of 3-PGA as a function of the reaction time for the RIMRs (solid red circles) and the BIMRs (open dark squares). The red dash-dotted line and dark dashed line are the liner fitting regressions. The slope of liner fitting regression of 3-PGA production in RIMRs is 13.8 $\mu mol\, g^{-1}$ RuBisCO $min^{-1}$ and that in BIMRs is almost 0. The initially injected RuBisCO and BSA concentrations for immobilization are both 6.25 $\mu g\, \mu L^{-1}$. The collected production solutions I from RIMRs and BIMRs are 21 $\mu L$. RuBP concentration is 0.5 mM. $HCO_3^-$ in the reaction buffer is 66 mM. Reaction temperature is 30 °C. **b** HPLC–MS/MS chromatography of RuBP and 3-PGA in the production solutions I obtained from RIMRs and BIMRs at the reaction time of 5 min. Error bars represent the standard deviations from three independent experiments. Source data are provided as a Source Data file

**Feasibility of RIMRs**. To check the feasibility of the reactors for artificial chemosynthesis of glucose precursor, the prepared RIMRs are used to produce 3-PGA from RuBP and $CO_2$ under different reaction times. In the control experiment that albumin from bovine serum (BSA) is immobilized onto the microfluidic reactor, the BSA-immobilized microfluidic reactors (BIMRs) produces negligible amount of 3-PGA (dark open squares in Fig. 2a), and the production amount does not increase with reaction time either. In contrast, the RIMRs demonstrate a linearly increased production of 3-PGA at the rate of 13.8 $\mu mol\, g^{-1}$ RuBisCO $min^{-1}$ (red solid circles and red dash-dotted fitting line of Fig. 2a) when the reaction time is 5 min or shorter. Production of 3-PGA by free RuBisCO as a function of time is provided in Supplementary Fig. 9 for comparison.

Apart from the UV–visible spectrometer, a high-performance liquid chromatography–tandem mass spectrometer (HPLC–MS/MS) system is also applied to directly examine the ingredients of 3-PGA and RuBP in the production solution I. The retention times of standard RuBP and 3-PGA are 16.97 and 15.21 min, respectively (see Supplementary Fig. 10a). The production amount of 3-PGA can also be derived from the calibration of the integrated peak areas as a function of its concentrations (see Supplementary Fig. 10b). As shown in Fig. 2b, the peaks with the similar retention time of RuBP and 3-PGA are observed in the production solution I obtained from the RIMRs. The 3-PGA production amount by RIMRs in 5 min is calculated to be about 71.9 $\mu mol\, g^{-1}$ RuBisCO from the chromatography. It is consistent with the 3-PGA production amount of 68.2 ± 5.4 $\mu mol\, g^{-1}$ RuBisCO in 5 min as determined by UV–visible spectrometer. However, no significant 3-PGA amount can be detected from the production solution I of the BIMRs. It is also noted that the RuBP signal from BIMRs is higher than that from RIMRs, indicating the conversion of RuBP to 3-PGA in RIMRs.

In consequence, it is well proved that 3-PGA is successfully synthesized from the RIMRs by implementing the light-

independent reactions pathway. Moreover, in the RIMRs reaction, the products are flowed through the reactor and pumped out immediately after reaction. The accumulation of 3-PGA around RuBisCO is reduced and the control of the reaction time can be achieved straightforwardly by changing the flow rate of the reactant mixture injection. This is one of the core features of RIMRs.

**Storage and thermal stabilities of RIMRs**. The storage and thermal stabilities of RIMRs are very important to practical applications. Regarding the storage stability (see Fig. 3a), the half-life (the storage time that 50% of the initial activity can be retained after the incubation in reaction buffer at 4 °C) is about 2.6 days for the free RuBisCO, and it is enhanced to be three times (8 days) after the immobilization. In addition, the storage stability of the immobilized RuBisCO is enhanced by 7.2 times compared with the free one (43% vs. 6% of the initial activity retention) after the incubation for 15 days. More characterizations of free RuBisCO under different incubation time including sodium dodecyl sulfate–polyacrylamide gel electrophoresis (SDS–PAGE) are provided in Supplementary Fig. 11.

Figure 3b shows the influence of temperature on the stability of RuBisCO. The highest activities ($A_{ht}$) of immobilized and free RuBisCO both appear at 30 °C. And the optimal incubation temperature range (i.e., the temperature range that over 98% of maximum activity is retained after incubation) of RuBisCO is expanded from 24–34 °C to 29–40 °C after the immobilization. It is also worth noting that the immobilized RuBisCO retains 67% of $A_{ht}$ at 70 °C, whereas the free RuBisCO retains only 10%, showing a difference of 6.7 times. When the immobilized and free RuBisCO are incubated at 50 °C for a long time up to 60 min (as shown in Supplementary Fig. 12), the immobilized RuBisCO retains 75% of $A_{ht}$ while the free RuBisCO retains only 46%. These show that the immobilization significantly enhances the

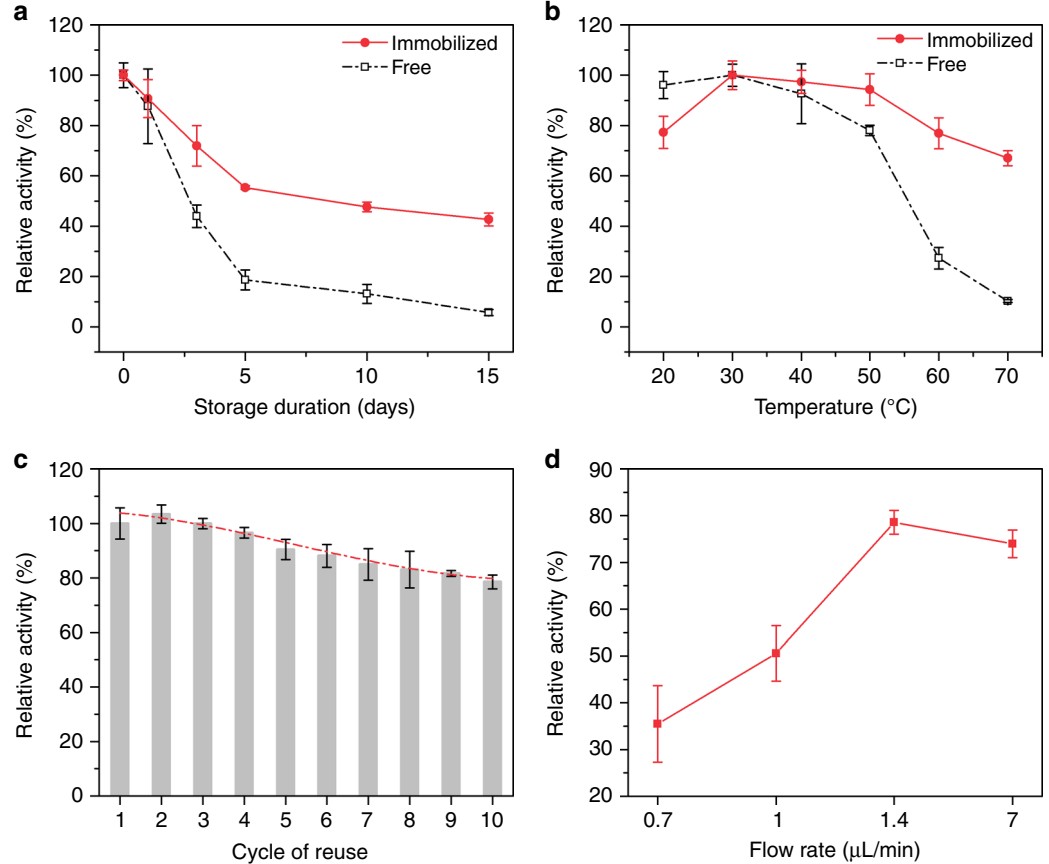

**Fig. 3** Stability and reusability of RIMRs. **a** Storage stability (incubation at 4 °C) of immobilized (red line) and free RuBisCO (dark line). **b** Thermal stability of immobilized (red line) and free RuBisCO (dark line). All samples are incubated for 10 min before the activity assay. **c** Reusability of the RIMRs when RuBP is injected at 1.4 μL min$^{-1}$ (the reaction time is 5 min). Red dash-dotted line is the third-order polynomial fitting, representing that the activity drop trend tends to slow down with the increase of cycles of reuse. **d** Relative activity as a function of the flow rate of RuBP injection (from 7 to 0.7 μL min$^{-1}$) for repeated uses. The amount of RuBisCO is 21.875 μg for both the immobilized and free ones in all the experiments. The volume of the collected production solutions for the storage and thermal stability tests are 100 μL. RuBP concentration is 0.5 mM and HCO$_3^-$ in the reaction buffer is 66 mM. Error bars represent the standard deviations from three independent experiments. Source data are provided as a Source Data file

**Table 2 Comparison of the RIMRs reaction and the bulk reaction**

| Reaction type | Storage stability[a] | Thermal stability[b] | Reusability[c] | Production of 3-PGA (μmol g$^{-1}$ RuBisCO) |
|---|---|---|---|---|
| RIMRs reaction | 43% | 67% | 90.4% | $\propto V_P^d$ |
| Bulk reaction | 6% | 10% | – | Saturation at ~360 |
| Enhancement factor | 7.2-fold | 6.7-fold | – | 1.8-fold[e] |

[a]Storage stability is defined as the remaining relative activity after 15 days
[b]Thermal stability is the remaining relative activity after incubation at 70 °C for 10 min
[c]Reusability is the remaining relative activity after five cycles of reuse at the flow rate of 1.4 μL min$^{-1}$
[d]$V_P$ is the volume of the reactant mixture (RuBP and HCO$_3^-$ in the reaction buffer)
[e]Data is obtained when $V_P$ is 1680 μL

resistance of RuBisCO to the thermal inactivation. For direct comparison, Table 2 lists the major performances of the immobilized and free RuBisCO.

Both the improved storage and thermal stabilities of RuBisCO can be ascribed to the enhanced stability of the enzyme conformation after the covalent immobilization[55,56]. Generally, the enzyme activity in solution could be greatly affected by the structure change caused by the unfolding of protein amino acid chains. On one hand, the elevated temperature or long-time storage would consequently influence the conformation of RuBisCO, leading to the activity reduction. On the other hand, the immobilization technique can offer multipoint anchors of RuBisCO on the support of the microfluidic reactors. As a result,

the enzyme structure and the active sites are properly preserved, thus maintaining the activity of immobilized RuBisCO.

**Reusability of RIMRs.** In addition to the thermal stability and the storage stability, enzymes are also required to have an outstanding reusability for industrial usage. As shown by the gray bars in Fig. 3c, the relative activity decreases with the increase of cycles of reuse but the trend of activity drop gradually slows down (red dash-dotted line by third-order polynomial fitting). More specifically, 90.4% and 78.5% of initial activity can be retained after five cycles and 10 cycles of reuse, respectively, when the flow rate is kept at 1.4 μL min$^{-1}$. The activity loss after long-time assay

may be attributed to two factors. The first factor is the enzyme inactivation after repeated uses and the inhibition of accumulative 3-PGA and RuBP degradation products in aqueous solution[50]. Therefore, RuBisCO activity is usually measured in a short assay that is stopped in <1 min[57]. The second factor is the enzyme detachment due to the flushing by the running flow. When the flow rate is lower down, the flow in the microchannel moves slower but the reaction time and the operation time is extended. Correspondingly, the flushing effect is reduced but the inactivation effect is increased, and their combination affects the reusability of RuBisCO in RIMRs. Based on the measured reusability at varying flow rates at 30 °C (see Fig. 3d and Supplementary Fig. 13), the flow rate of 1.4 μL min$^{-1}$ yields the highest remaining activity after 10 cycles of reuse. The reusability of RuBisCO in the bulk reaction is hard to determine, since it needs special techniques to filter out and recycle the enzyme. In contrast, it is quite facile in the RIMRs reaction by pumping out the production solution and then injecting the new reactant mixture.

**Continuous production of 3-PGA from RIMRs.** When the reactant mixture is pumped constantly into the RIMRs, continuous and accumulative production of 3-PGA at maximum efficiency can be achieved using only a small amount of RuBisCO. As shown in Fig. 4, when the volume of collected production solution ($V_p$) is small, the steric hindrance probably causes the RIMRs reaction to have a lower 3-PGA production than the bulk reaction. Nevertheless, the 3-PGA production by the RIMRs in the continuous mode (reaction temperature: 30 °C, reaction time: 1 min) increases with $V_p$ at a rate of 0.39 μmol g$^{-1}$ RuBisCO min$^{-1}$ L$^{-1}$ (red solid circles in Fig. 4). In contrast, the 3-PGA production in the bulk reaction goes up gradually and tends to saturate at about 360 μmol g$^{-1}$ RuBisCO when $V_p$ is larger than 600 μL (dark open squares in Fig. 4). The saturation of 3-PGA production in the bulk reaction is attributed to the reduced RuBisCO concentration with the increased $V_p$ when the amount of RuBisCO remains the same. Regarding the RIMRs reaction, the increase of $V_p$ does not affect the RuBisCO concentration thanks to the immediate pumping out of the production solution I and the continuous injection of fresh reactant mixture, which also help to avoid the feedback inhibition.

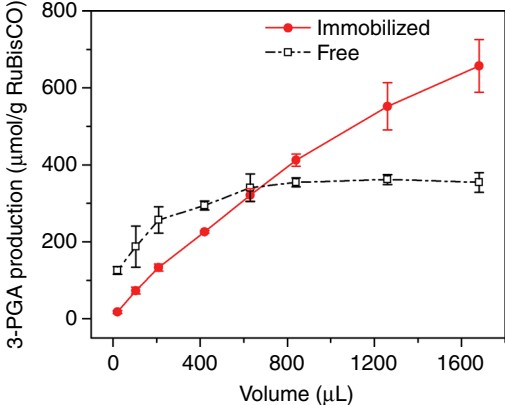

**Fig. 4** Continuous production of 3-PGA as a function of the volume of collected production solution. The dark open squares represent the 3-PGA production in the bulk reaction with free RuBisCO and the red solid circles represent the 3-PGA production in the RIMRs with the immobilized RuBisCO. The amount of RuBisCO used is 21.875 μg for both the immobilized and free ones. RuBP concentration is 0.5 mM and HCO$_3^-$ in the reaction buffer is 66 mM. Reaction temperature is 30 °C. Reaction time is 1 min. Error bars represent the standard deviations from three independent experiments. Source data are provided as a Source Data file

Together with the good reusability of immobilized RuBisCO, the RIMRs enable the accumulative production of 3-PGA using only a small amount of RuBisCO. This is another core merit of RIMRs.

## Discussion

In this work, we have immobilized RuBisCO into the PDA-modified microfluidic reactors for continuous production of glucose precursor. As summarized in Table 2, the immobilized RuBisCO showed a 7.2-fold improvement in the storage stability and a 6.7-fold increase in the thermal stability compared with the free RuBisCO. The RIMRs also presented a good reusability with 90.4% of its initial activity retained after five cycles of reuse and 78.5% after 10 cycles. This work lays the foundation to the continuous synthesis of glucose precursor by building up the NPS pathway in the bio-mimic microfluidic reactors. Many advantages of microfluidics for enzymatic reactions are perfectly demonstrated here. Compared with the bulk system, the reaction in microfluidic reactor is easy to control by adjusting the injection of reactant mixture solution. It can also continuously produce 3-PGA by constantly injecting CO$_2$ and RuBP, enabling the accumulative production of 3-PGA with very small amount of RuBisCO in the continuous mode. Moreover, the continuous pumping out of products and injecting in the reactant mixture avoid the feedback inhibition problem and thus maximize the efficiency of RuBisCO enzymatic reaction in a low-cost, convenient, and durable way.

As observed in the experiment, the RuBisCO immobilization would cause some loss of activity due to the detachment and the inactivation of enzyme. In the future, we will try other immobilization methods, like the enzyme encapsulation[45,58], with the hope to better retain the activity. Moreover, the immobilization amount of RuBisCO is still small due to the limited space of the microreactor. Some new material such as MOFs[59], COFs[60], mesoporous nanoparticles[61], and carbon nitride[62] could be applied to the microreactor to provide larger surface areas in the future. The reactor can also be easily scaled up[63–66] to increase the output and more functionalities can be integrated, such as deoxygenation, temperature control of individual reaction, etc. Other enzymes could be incorporated into the microreactor to cascade the reactions for the constitution of full NPS pathway. In a long run, a scientific approach would be available to produce massive basic food materials to relieve the food crisis and to prepare for future space colonization.

## Methods

**Chemicals and reagents.** The reaction buffer (pH 8.0) used for enzyme assay consisted of 0.1 M tris(hydroxymethyl)aminomethane buffer (Tris–HCl, pH 8.0, Beijing Solarbio Technology Co., Ltd., Beijing), 5 mM magnesium chloride hexahydrate (MgCl$_2$·6H$_2$O, AR, Sinopharm Chemical Reagent Co., Ltd., Shanghai), 66 mM potassium bicarbonate (KHCO$_3$, AR, Sinopharm Chemical Reagent Co., Ltd, Shanghai), and 5 mM DL-dithiothreitol (DTT, 99%, Aladdin Industrial Corporation, Shanghai). (3-aminopropyl)triethoxysilane (APTES, 99%) was bought from Aladdin Industrial Corporation, Shanghai. Dopamine hydrochloride (98%, J&K Scientific Ltd., Beijing) was used as 1 mg mL$^{-1}$ solution in 10 mM Tris–HCl buffer (pH = 8.8, Beijing Solarbio Technology Co., Ltd., Beijing). Reduced nicotinamide adenine dinucleotide disodium salt (NADH Na$_2$, ≥98.0%), adenosine-5′-triphosphate disodium salt trihydrate (ATP Na$_2$, ≥98.0%) and fluorescein isothiocyanate isomer I (FITC, ≥90%, HPLC) were purchased from Beijing Solarbio Technology Co., Ltd. Albumin (98%, from bovine serum) was from J&K Scientific Ltd., Beijing. D-ribulose 1,5-bisphosphate sodium salt hydrate (RuBP, ~90%), D-ribulose 1,5-diphosphate carboxylase (RuBisCO, from spinach partially purified powder, 0.01–0.1 unit/mg solid), D-(−)−3-phosphoglyceric acid disodium salt (3-PGA, ≥93%), glyceraldehyde 3-phosphate dehydrogenase (GAPDH, from rabbit muscle lyophilized powder), 3-phosphoglyceric phosphokinase (PGK, from baker's yeast (*S. cerevisiae*), ammonium sulfate suspension, ≥1000 units/mg protein), glycerol 3-phosphate oxidase (G3POX, from *Pediococcus* sp. lyophilized powder), α-glycerophosphate dehydrogenase (G3PDH, from rabbit muscle, type I, ammonium sulfate suspension), triosephosphate isomerase (TPI, from baker's yeast (*S. cerevisiae*), ammonium sulfate suspension) and catalase (from bovine liver powder) were provided by Sigma-Aldrich.

**Fabrication of RIMRs**. The RIMRs were made in three main steps: (a) preparation of the pristine polydimethylsiloxane (PDMS) microfluidic reactors; (b) modification of PDA; and (c) immobilization of RuBisCO. First, the PDMS microfluidic reactors were fabricated by sealing one molded PDMS layer with another flat PDMS layer on the top. The molded PDMS layer was made by using the standard soft-lithography technique[67] with the Sylgard 184 elastomer kit (Dow Corning Corporation) and a SU-8 mold (SU-8 50, MicroChem) fabricated by the photo-lithography. Then, the pristine PDMS inner surface of the microfluidic reactors were modified by PDA. The strategy for the PDA modification was adopted from Zheng's group with minor modification[38]. The detailed procedures are provided in Supplementary Methods. Next, 7 µL of RuBisCO in the reaction buffer was injected into the PDA-PDMS microfluidic reactor by pipette and kept at room temperature for 6 h for the RuBisCO immobilization. After that, the reactor was rinsed by the reaction buffer using a syringe pump at a flow rate of 2.5 µL min$^{-1}$ for 40 min. Finally, the RIMRs were ready for use.

**Confirmation of RuBisCO immobilization**. The surface characterization of the inner surfaces of the microfluidic reactors after each step of RuBisCO immobilization was first conducted by scanning electron microscope (SEM, JEOL JSM-6490) for the confirmation of RuBisCO immobilization. More characterization details such as optical microscopic images, water contact angles, Raman spectra, ATR-FTIR spectra and fluorescence images are presented in Supplementary Figs. 2–6.

**Determination of protein-loading capacity**. The protein-loading capacity of the PDA-modified microfluidic reactors was investigated to select an optimal RuBisCO concentration for the RIMRs fabrication in the following research. Increased concentrations of RuBisCO solutions (1.5625–25 µg µL$^{-1}$) were injected into the PDA-modified microfluidic reactors. Then, the rinsed-out RuBisCO solutions were carefully collected to determine the protein concentration difference between the initially injected and the rinsed-out RuBisCO solutions. The protein concentration was qualified by the Bradford method[68] using the Quick Start Bradford Protein Assay kit (Bio-Rad Pacific Limited.), which were determined by measuring the absorbance at the wavelength of 595 nm using a UV–visible spectrometer (UV-2450, Shimadzu). BSA solutions (0.125–1 mg mL$^{-1}$) were selected as standards to plot the calibration curve (see Supplementary Fig. 7). The protein-loading amount of the reactor was calculated by

$$\text{protein} - \text{loading amount}(\mu g \cdot cm^{-2}) = (C_0 V_i - C_1 V_w)/A_r \quad (1)$$

where $C_0$ is the protein concentration of initially injected RuBisCO solution (µg µL$^{-1}$), $C_1$ is the protein concentration of the rinsed-out RuBisCO solution (µg µL$^{-1}$), $V_i$ is the volume of initially injected RuBisCO solution (µL), $V_w$ is the volume of washing solution collected from the outlet of the reactor (µL), and $A_r$ is the surface area of the reactor (~367.5 mm$^2$). The maximum protein-loading amount was then regarded as the protein-loading capacity of the PDA-modified microfluidic reactor. The protein-loading efficiency was also obtained by

$$\text{protein} - \text{loading efficiency}(\%) = (C_0 V_i - C_1 V_w)/C_0 V_i \quad (.2)$$

Then the optimal RuBisCO concentration for further research can be determined referring to the protein-loading capacity and the protein-loading efficiency. All the error bars represent the standard deviations from three repeated experiments.

**Assay of RuBisCO activity**. The activities of the free and immobilized RuBisCO were determined using an amplification signal assay adapted from the previously reported method[57]. The general details are provided in Supplementary Methods. In the immobilized RuBisCO activity assay, reactant mixture (66 mM HCO$_3^-$ and 0.5 mM RuBP in the reaction buffer) was passed through the RIMRs at the flow rate of 7 µL min$^{-1}$ (reaction time is 1 min). The production solution I (mixture of RuBisCO, RuBP, HCO$_3^-$ and products) was collected from the outlet of the reactors before it was added into the assay mixture for 3-PGA amount determination. The specific composition of the assay mixture can be seen in Supplementary Methods of amplification method for RuBisCO activity assay. For the free RuBisCO activity assay, RuBisCO was incubated with reactant mixture for 1 min before the reaction is stopped by the same volume of 80% ethanol. Then, the production solution II (mixture of RuBisCO, RuBP, HCO$_3^-$, products, and ethanol) was added into the assay mixture to determine the 3-PGA amount. The amount of RuBisCO used here was the same as that immobilized into the microfluidic reactors (21.875 µg). The amount of 3-PGA in the production solutions could be determined from the calibration lines generated by adding different amounts of standard 3-PGA into the assay mixture (see Supplementary Fig. 8). Here the calibration line for production solution I is obtained by using standard 3-PGA dissolved in the reaction buffer, and that for production solution II is prepared by using standard 3-PGA dissolved in the reaction buffer:ethanol (60:40, v/v). The enzyme activity was defined as the production rate of 3-PGA (µmol g$^{-1}$ RuBisCO min$^{-1}$). The RuBisCO activities under different concentrations of RuBP were fit to a Michaelis–Menten-type model using the hyperbola regression to derive the $V_{max}$ and $K_m$ parameters of both free and immobilized RuBisCO.

**Feasibility of 3-PGA production using RIMRs**. The feasibility of producing 3-PGA with RIMRs was tested by injecting 0.5 mM RuBP and 66 mM HCO$_3^-$ in the reaction buffer into the as-prepared RIMRs using a syringe pump at a specific flow rate. The production solution I was collected from the outlet for the 3-PGA amount determination by the amplification signal assay. Different amounts of 3-PGA can be produced when the reactant mixture is injected at different flow rates (from 28 to 1.4 µL min$^{-1}$), which directly determines the reaction time of the 3-PGA production reaction (see Supplementary Methods for reaction time determination). The produced amount of 3-PGA (µmol g$^{-1}$ RuBisCO) in a specific reaction time can also be determined by the same standard curve obtained from last section. Control experiments were conducted at the same time with the same amount of BSA that was immobilized in microfluidic reactors. HPLC–MS/MS was also applied to detect the produced 3-PGA from the production solution I according to the method tested before[69]. The details of HPLC–MS/MS analysis are described in the Supplementary Methods.

**Storage and thermal stabilities of RIMRs**. To test the storage stability of the RIMRs, several prepared RIMRs were incubated at 4 °C for different days. Then, the immobilized RuBisCO activities of them were measured. The storage stability of free RuBisCO was also tested at the same condition for comparison. The highest RuBisCO activity ($A_{hs}$) was normalized to 100% and the relative RuBisCO activities after different storage days were calculated as a percentage of $A_{hs}$.

Regarding the thermal stability of the RIMRs, several prepared RIMRs were first incubated in the oven at different temperatures (from 20 to 70 °C) for 10 min and then the RuBisCO activities of each RIMRs were examined. RIMRs were also incubated at 50 °C for up to 60 min to examine their stability at elevated temperature for a prolonged time. The thermal stability of free RuBisCO was also tested at the same condition for comparison. $A_{ht}$ was normalized to 100% and the relative RuBisCO activities at different incubation temperatures were calculated as a percentage of $A_{ht}$.

**Reusability of RIMRs**. The reusability of the RIMRs means the capability of the RIMRs being used repeatedly. It was evaluated by conducting the 3-PGA production reaction using one reactor for 10 cycles of reuse. Here, one cycle of reuse refers to the test that 21 µL of reactant mixture solution passes through the RIMRs for 3-PGA production. RuBisCO activity was measured from the collected production solution I for each cycle of reuse. Different flow rates were applied to inject the reactant mixture to investigate the effects of flushing. The relative RuBisCO activities were calculated as a percentage of the initial RuBisCO activity in the first cycle.

**Continuous production of 3-PGA from RIMRs**. Continuous production of 3-PGA was achieved by constantly injecting the reactant mixture into the RIMRs and thus a large amount of 3-PGA can be produced. As a comparison, 3-PGA production in the bulk reaction with the same amount of free RuBisCO was conducted at the same time. Increasing $V_p$ (from 21 to 1680 µL) was applied to examine the difference of the 3-PGA production ability between the RIMRs reaction and the bulk reaction.

**Reporting summary**. Further information on research design is available in the Nature Research Reporting Summary linked to this article.

## Data availability
The source data underlying Fig. 1d, Table 1, Figs. 2a, b, 3a–d, 4, and Supplementary Figs. 4–10, 11b, 12, 13 are provided as a Source Data file. All data underlying the findings in this study are available from the corresponding author upon reasonable request.

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

## Acknowledgements

This work is supported by National Science Foundation of China (no. 61377068) and Research Grants Council (RGC) of Hong Kong (N_PolyU505/13, 152184/15E, 152127/17E, 152126/18E, and 152219/19E). The authors would like to thank The Hong Kong Polytechnic University for the grants G-YBPR, 4-BCAL, 1-ZE14, 1-ZE27, and 1-ZVGH, Shandong Provincial Natural Science Foundation, China (ZR2016XJ004) and Doctoral cooperation fund of Shandong Academy of Sciences (2017BSHZ007). The technical assistance in mass spectrometry facility of the University Research Facility in Life Sciences (ULS) of the Hong Kong Polytechnic University is also appreciated.

## Author contributions

Y.Z. designed the research, performed the experiments and data analysis, and prepared the manuscript. Z.H. and X.H. contributed to UV experiments and manuscript revision. Q.C., L.S. and Y.Y. contributed to designing and fabricating the microfluidic reactors. Q.W., P.-K.S. and Z.Y. contributed to the HPLC–MS/MS experiments. Y.J. guided the biotreatment, Z.L. helped the analysis, W.Y. aided the surface modification, A.J., S.S., and W.Z. supported the spectral assay. X.Z. conceived the idea, supervised the whole project and revised the manuscript. All authors discussed the results and commented on the manuscript.

## Additional information

**Competing interests:** The authors declare no competing interests.

