## [Peer Review File · Nature Communications]

Reviewers' comments:

Reviewer #1 (Remarks to the Author):

Ribulose-1,5-bisphosphate carboxylase/oxygenase (RuBisCO) is arguably one of the most abundant proteins in the biosphere and a key enzyme in the global carbon cycle. In fact, more than 90% of the inorganic carbon that is converted into biomass is fixed by the enzyme RuBisCO that catalyzes the carboxylation and cleavage of ribulose-1,5-bisphosphate (RuBP) into 3-phosphoglycerate (3PG). Unfortunately, RuBisCO suffers from low catalytic rate and poor specificity.

In the present work, RuBisCO was covalently immobilized into a microfluidic reactor and used for the production of 3PG. The microreactor showed enhanced enzyme stabilities (both under storage and thermal stability) and reusability compared to the free form. The strength of the work is represented by being the first application of immobilised RuBisCO in microreactors.

The manuscript is well written and the reported data are technically sound and precisely reported and discussed. I would like to recommend publication of this work in Nature Communication considering the following issues:

- In the abstract and in the Discussion the Authors state that this work "paves the way to replicating the whole light-independent reactions of NPS to produce large amount of basic food materials for food crisis relief and future space colonization" (lines 14-15) and that it represents "a promising industrial route to produce large amount of basic food material is accordingly offered for the food crisis relief and future space colonization" (lines 258-260). These statements need to be softened. At present, they are reporting a microreactor able to work at few microliters/minutes. The output is far away to be interesting for industrial applications. Otherwise, they could report a study of the scalability and product isolation as they state they will do in future.
- The Authors reports the stability of the enzyme after 5 cycles. This is a quite limited number of cycles. What happens after 10 or more cycles using the lower flow rate? This data should be added

Reviewer #2 (Remarks to the Author):

In this manuscript the enzyme RuBisCO was covalently immobilized into the channel of a polydimethylsiloxane microreactor and the resulting properties were studied in different experimental conditions.

The idea to prepare microfluidic reactors, in analogy with the natural distribution of RuBisCO in chloroplast, is interesting and well realized, but the authors seem to underestimate the problems deriving from the use of micro channels in real practical application. Moreover some specific point should be addressed and some experiments improved before consider this manuscript suitable for publication.

Remarks:

- Lines 49-53 and 57-59: In my opinion, not only the advantages, but also the problems deriving from the immobilization procedures and the use of a microreactor should be considered to give a correct evaluation of these techniques.
- Line 106 and Fig S2b: FTIR analysis evidences the presence in the RIMRs spectrum of a weak band at about 1600 cm⁻¹ that could be referable to the vibrational modes of the peptide bond (amide bands). I suggest to better consider this absorption, since it would be both a confirm of the effective RuBisCO immobilization and give information about possible conformational changes induced by the immobilization procedure.
- Lines 149-150: It is not clear to me what the authors meant with the sentence "the reactant may have an increased diffusion length to the enzyme". Please explain the concept and/or modify the sentence.
- Lines 172-174 and Fig 6b: Although HPLC measurements require a calibration curve to be used quantitatively, the area of the RuBP peak from RIMRs should appear clearly lower, compared to that obtained from BIMRs, as in this case the reaction led to the formation of 3-PGA. Why the RuBP peaks from RIMRs and BIMRs are identical?
- Lines 190-197 and Fig 7b: The authors incubated the samples in oven for 10 mins (line 368) in order to test their thermal stability, but I think that this kind of experiment is not informative of

the real stability of the RIMRs. In my opinion, two different test should be made to evaluate the possibility to use the RIMRs in 3-PGA production: the first with incubation time coherent with the retention time of the substrate in the microreactor (depending on the chosen flux), and the second with a prolonged use of the RIMRs for the continuous production of 3-PGA.

- Line 246: I do not agree with authors' affirmation about the "excellent reusability" of RIMRs.

Their data about RuBisCO activity after the immobilization on the micro reactors are good, but a 10% decrease of enzymatic activity after only 5 reaction cycle is not sufficient to hypothesize a real use of RIMRs. More data are needed in order to understand if the slightly negative trend shown in Figure 7c could accelerate or slow down after other reaction cycles.

- Lines 258-259: The sentence "A promising industrial route to produce large amount of basic food material" it does not seem appropriate to summarize these results if we consider the dimensions of the microreactor respect to the amount of food requested to resolve "the food crisis relief and future space colonization".

- Lines 324-325: How was the surface area of the microreactor determined?

- All the text: Some errors are present. Please edit the manuscript and check it for typo errors.

Reviewer #3 (Remarks to the Author):

This manuscript describes the immobilization of a protein onto a solid surface by amine coupling followed by characterization of its activity and various properties such as thermal stability. This is a very routine type of experiment and thus is not well suited for a non-specialty journal. I believe there are engineering journals that will publish this type of analysis.

The authors purport to be testing some type of artificial photosynthesis, but I don't think this is an accurate description. If you add RuBP and CO₂ to Rubisco you will get 3PG and this is not a recent discovery. In the same vein you could perform any other reaction in the Calvin cycle and claim it to be APS. In fact the Calvin cycle is independent of light, often being powered chemically, it would more accurately be described as artificial chemosynthesis.

A key problem is already seen in the title : "continuous artificial synthesis of glucose precursor from CO₂". This is wrong – shown is synthesis of glucose precursor from CO₂ and ribulose 1,5-bisphosphate- a high energy compound that is purchased at the cost of ~\$1000/ 100 mg from Sigma, which indicates that feeding the world using this concept would be quite challenging.

To claim APS it is necessary to have light and then obtain some type of energy storage chemical. I don't see how it is sufficient to demonstrate that one is able to replicate an assay of the world's best studied enzyme.

I could envisage work in this vein to become interesting for Nature Communications if the current title was in fact accurate – conversion of CO₂ to G3P- not assaying a single exceedingly well studied enzymatic reaction. However there are clearly many issues that make this challenging.

Major points:

1- Regarding Figs. 2-5 and Results p. 5-7

A large portion of this manuscript describes the immobilization of a protein on a solid surface. This is a trivial procedure, routinely performed in biophysics in applications such as surface plasmon resonance and does not warrant significant characterization to be described in the main text of a high impact publication. Such characterization should however be carefully described in the supplementary information and occupy at most 2-3 sentences in the main manuscript. However the presented characterization is only meaningful if it is also performed for a control- hence the same should be done for the BSA immobilization (that presumably will result in identical data?)

2- Table 1: the reported V_{max} when converted to turnover per active site for the bulk reaction is equivalent to ~3 min⁻¹ – however Spinach rubisco has a k_{cat} of about 2s⁻¹ , this suggests the enzyme that is being used here (powder purchased from sigma) is essentially non-functional. . maybe there was a calculation error somewhere? Alternatively the unusual (relying on signal amplification rather than a direct readout) spectrophotometric assay used here may not have been implemented successfully. This needs to be carefully checked.

One issue to check carefully concerns the inclusion of significant amounts of ethanol (used to stop the reaction) in the coupled assay- maybe this is responsible for the very low activities reported. It would help to present the original data used to derive the reported activities in the supplementary information (not just the 3PG calibration curve).

Related to this the activity reported in Fig. 6a is also incredibly slow (inactive) with one active site producing 0.5 molecules of 3PG per minute..

The authors will also need to consider the following points when submitting future versions of this manuscript:

1- Pg. 2 "we also carried out a series of studies on the light-dependent reactions... very high efficiency" – this is a strange claim in the absence of a reference. Either remove or provide citation.

2- Fig. 1b- the chemical structures of amine linked Rubisco to the surface does not look correct to me (Nitrogen bonding covalently to the center of an aromatic ring)- please check

3- The legend to Fig. 7a does not indicate the temperature used in this experiment. (the methods says 4 deg C – you also need to define concentration and buffer conditions)- the loss of activity is in fact concerning, but should not be observed.. Rubisco is a very stable enzyme. This should be characterized more carefully- does migration behaviour on a native gel change over time? Is degradation observed on SDS-PAGE? Does the protein precipitate? The loss of activity may suggest the enzyme preparation (from Sigma) is still quite impure.

4- Related to above- define protein concentration and buffer conditions for every experiment- also the thermal stability assay.

5- Fig. 7b does not explain the experiment in the legend.

Response to Reviewer 1

General comment: Ribulose-1,5-bisphosphate carboxylase/oxygenase (RuBisCO) is arguably one of the most abundant proteins in the biosphere and a key enzyme in the global carbon cycle. In fact, more than 90% of the inorganic carbon that is converted into biomass is fixed by the enzyme RuBisCO that catalyzes the carboxylation and cleavage of ribulose-1,5-bisphosphate (RuBP) into 3-phosphoglycerate (3PG). Unfortunately, RuBisCO suffers from low catalytic rate and poor specificity.

In the present work, RuBisCO was covalently immobilized into a microfluidic reactor and used for the production of 3PG. The microreactor showed enhanced enzyme stabilities (both under storage and thermal stability) and reusability compared to the free form. The strength of the work is represented by being the first application of immobilized RuBisCO in microreactors.

Response: Sincere appreciation to the Reviewer for the interest in this work and the supportive comments. Indeed, the originality of this work lies mostly in the first try of replicating the first part of light-independent reactions of photosynthesis of glucose by immobilizing RuBisCO in microreactors.

Comment 1: In the abstract and in the Discussion the Authors state that this work “paves the way to replicating the whole light-independent reactions of NPS to produce large amount of basic food materials for food crisis relief and future space colonization” (lines 14-15) and that it represents “a promising industrial route to produce large amount of basic food material is accordingly offered for the food crisis relief and future space colonization” (lines 258-260). These statements need to be softened. At present, they are reporting a microreactor able to work at few microliters/minutes. The output is far away to be interesting for industrial applications. Otherwise, they could report a study of the scalability and product isolation as they state they will do in future.

Response 1: We agree with the comment that it is a bit too far for this work to jump directly to real industrial production. In the revised manuscript, we have softened the statement as below.

This work, though still far from industrial production, makes the first move towards replicating the whole light-independent reactions of NPS to artificially produce basic food materials, which may hold the key to food crisis relief and future space colonization.

--- Abstract, the last 4 lines

The reactor can also be easily scaled up⁵⁴⁻⁵⁷ to increase the output and more functionalities can be integrated, such as deoxygenation, temperature control of individual reaction, etc.

--- Page 15, paragraph 2, lines 277 – 278

In a long run, a scientific approach would be available to produce massive basic food materials to relieve the food crisis and to prepare for future space colonization.

--- Page 15, paragraph 2, the last 2 lines

Comment 2: The Authors reports the stability of the enzyme after 5 cycles. This is a quite limited number of cycles. What happens after 10 or more cycles using the lower flow rate? This data should be added

Response 2: Following the Reviewer's comment, we have tested the stability of immobilized RuBisCO for 10 cycles at different flow rates and have updated the data to Fig. 3c & d in the revised manuscript and Fig. S13 in the revised Supplementary Information. The general trend is that the activity keeps decreasing after more cycles as shown in Fig. 3c. Nevertheless, 78.5% of initial activity can be maintained after ten cycles of reuse at the flow rate of $1.4 \mu\text{L}\cdot\text{min}^{-1}$ (the corresponding reaction time is 5 min per cycle). The stability is apparently affected by the flow rate as plotted in in Fig. 3d and Fig. S13. The remaining activity after 10 cycles follows, from high to low, the order of $1.4 \mu\text{L}\cdot\text{min}^{-1}$, $7 \mu\text{L}\cdot\text{min}^{-1}$, $1 \mu\text{L}\cdot\text{min}^{-1}$ and $0.7 \mu\text{L}\cdot\text{min}^{-1}$. The higher remaining activity at $1.4 \mu\text{L}\cdot\text{min}^{-1}$ than that at $7 \mu\text{L}\cdot\text{min}^{-1}$ may be due to the fact that a lower flow rate results in a lower chance for RuBisCO being flushed off from the support. However, a flow rate lower than $1.4 \mu\text{L}\cdot\text{min}^{-1}$ does not lead to a larger remaining activity. Apart from the flushing-off effect, this can also be ascribed to the fall over of activity after a longer operation time. The total operation times for 10 cycles are 210 min and 300 min for $1 \mu\text{L}\cdot\text{min}^{-1}$ and $0.7 \mu\text{L}\cdot\text{min}^{-1}$, respectively, which are 1.4-fold and 2-fold of the total operation time 150 min for $1.4 \mu\text{L}\cdot\text{min}^{-1}$. Detailed descriptions are added to the revised manuscript as listed below.

It also exhibited a good reusability with 90.4% activity retained after 5 cycles of reuse and 78.5% after 10 cycles.

--- Abstract, lines 11 – 12

The RIMRs also presented a good reusability with 90.4% of its initial activity retained after 5 cycles of reuse and 78.5% after 10 cycles.

--- Page 14, lines 260 – 261

Fig. 3 **c** Reusability of the RIMRs when RuBP is injected at $1.4 \mu\text{L}\cdot\text{min}^{-1}$ (the reaction time is 5 min). Red dash-dotted line is the third-order polynomial fitting, representing that the activity drop trend tends to slow down with the increase of cycles of reuse. **d** Relative activity as a function of the flow rate of RuBP injection (from $7 \mu\text{L}\cdot\text{min}^{-1}$ to $0.7 \mu\text{L}\cdot\text{min}^{-1}$) for repeated uses.

As shown by the grey bars in Fig. 3c, the relative activity decreases with the increase of cycles of reuse but the trend of activity drop gradually slows down (red dash-dotted line by third-order polynomial fitting). More specifically, 90.4% and 78.5% of initial activity can be retained after 5 cycles and 10 cycles of reuse, respectively, when the flow rate is kept at $1.4 \mu\text{L}\cdot\text{min}^{-1}$. The activity loss after long-time assay may be attributed to two factors. The first factor is the enzyme inactivation after repeated uses and the inhibition of accumulative 3-PGA and RuBP degradation products in aqueous solution⁴⁵. Therefore, RuBisCO activity is usually measured in a short assay that is stopped in less than 1 min⁴⁸. The second factor is the enzyme detachment due to the flushing by the running flow. When the flow rate is lower down, the flow in the microchannel moves slower but the reaction time and the operation time is extended. Correspondingly, the flushing effect is reduced but the inactivation effect is increased, and their combination affects the reusability of RuBisCO in RIMRs. Based on the measured reusability at varying flow rates at 30°C (see Fig. 3d and Fig. S13), the flow rate of $1.4 \mu\text{L}\cdot\text{min}^{-1}$ yields the highest remaining activity after 10 cycles of reuse. The reusability of RuBisCO in the bulk reaction is hard to determine since it needs special techniques to filter out and recycle the enzyme. In contrast, it is quite facile in the RIMRs reaction by pumping out the production solution and then injecting the new reactant mixture.

--- Page 12, paragraph 2, lines 217 – 233

The reusability of RIMRs was evaluated by injecting the reactant mixture into the reactor for ten cycles of reuse at different flow rates. The reusability of RuBisCO at different flow rates are plotted in Fig. S13a. The relative activities after different cycles of reuse are plotted as a function of flow rate in Fig. S13b.

Fig. S13 Reusability of the RIMRs for different cycles of reuse when the reactant mixture is injected at the flow rates of $7 \mu\text{L}\cdot\text{min}^{-1}$, $1.4 \mu\text{L}\cdot\text{min}^{-1}$, $1 \mu\text{L}\cdot\text{min}^{-1}$ and $0.7 \mu\text{L}\cdot\text{min}^{-1}$. As the volume of reactor is $7 \mu\text{L}$, the corresponding reaction time is 1 min, 5 min, 7 min and 10 min, respectively. (a) Reusability as a function of cycle of reuse at different flow rates. (b) Reusability as a function of flow rate after different cycles of reuse.

--- Page S14, session 10 of Supplementary Information

Response to Reviewer 2

General comment: In this manuscript the enzyme RuBisCO was covalently immobilized into the channel of a polydimethylsiloxane microreactor and the resulting properties were studied in different experimental conditions.

The idea to prepare microfluidic reactors, in analogy with the natural distribution of RuBisCO in chloroplast, is interesting and well realized, but the authors seem to underestimate the problems deriving from the use of micro channels in real practical application. Moreover some specific point should be addressed and some experiments improved before consider this manuscript suitable for publication.

Response: Sincere appreciation to the Reviewer for the interest in this work and the supportive comments. Indeed, the originality of this work lies mostly in the first try of replicating the first part of light-independent reactions of photosynthesis of glucose by immobilizing RuBisCO in microreactors. For the other points and the experiment improvements, they will be presented in the responses to the following comments. For the problems of microchannels, we have followed the Reviewer's comment to discuss them in the revised manuscript (see below).

This work, though still far from industrial production, makes the first move towards replicating the whole light-independent reactions of NPS to artificially produce basic food materials, which may hold the key to food crisis relief and future space colonization.

--- Abstract, the last 4 lines

Some new material such as MOFs⁵⁰, COFs⁵¹, mesoporous nanoparticles⁵², and carbon nitride⁵³ could be applied to the microreactor to provide larger surface areas in the future. The reactor can also be easily scaled up⁵⁴⁻⁵⁷ to increase the output and more functionalities can be integrated, such as deoxygenation, temperature control of individual reaction, etc.

--- Page 15, paragraph 2, lines 275 – 278

Comment 1: Lines 49-53 and 57-59: In my opinion, not only the advantages, but also the problems deriving from the immobilization procedures and the use of a microreactor should be considered to give a correct evaluation of these techniques.

Response 1: We indeed find some problems during the experiments and agree with this comment that the problems should be discussed, for example, the immobilization would cause some loss of enzyme activity, and the microreactor has limited surface for enzyme attachment and low process throughput. For these problems, they are discussed as follows in the revised manuscript.

Of course, nothing is perfect. The immobilization may cause some reduction of enzyme activity due to detachment and inactivation, which will be studied in this work as well.

--- Page 3, paragraph 1, lines 62 – 63

As observed in the experiment, the RuBisCO immobilization would cause some loss of activity due to the detachment and the inactivation of enzyme. In the future, we will try other immobilization methods, like the enzyme encapsulation^{35,49}, with the hope to better retain the activity. Moreover, the immobilization amount of RuBisCO is still small due to the limited space of the microreactor. Some new material such as MOFs⁵⁰, COFs⁵¹, mesoporous nanoparticles⁵², and carbon nitride⁵³ could be applied to the microreactor to provide larger surface areas in the future. The reactor can also be easily scaled up⁵⁴⁻⁵⁷ to increase the output and more functionalities can be integrated, such as deoxygenation, temperature control of individual reaction, etc.

--- Page 14, paragraph 2, lines 271 – 278

Comment 2: Line 106 and Fig S2b: FTIR analysis evidences the presence in the RIMRs spectrum of a weak band at about 1600 cm⁻¹ that could be referable to the vibrational modes of

the peptide bond (amide bands). I suggest to better consider this absorption, since it would be both a confirm of the effective RuBisCO immobilization and give information about possible conformational changes induced by the immobilization procedure.

Response 2: Grateful to the Reviewer for pointing out this important feature. Relative descriptions have been added to the revised manuscript and Supporting Information.

The activity loss induced by the immobilization procedure is ascribed to the conformational changes of RuBisCO. Such conformational changes can also be observed from the presence of a weak band at about 1600 cm^{-1} of RuBisCO-PDA-PDMS in Fig. S5b.

--- Page 7, paragraph 1, lines 126 – 129

Particularly, the FTIR spectrum of the RIMRs has a weak band at about 1600 cm^{-1} , which could be referred to the vibrational modes of the peptide bond (amide bands). This absorption band confirms the effective RuBisCO immobilization and gives information about conformational changes induced by the immobilization procedure.

--- Page S5, session 2.4 of Supporting Information

Comment 3: Lines 149-150: It is not clear to me what the authors meant with the sentence “the reactant may have an increased diffusion length to the enzyme”. Please explain the concept and/or modify the sentence.

Response 3: Sorry about the unclear description. In the bulk reaction, the reactant contacts and reacts with the free enzyme dispersed in the solution, the diffusion distance is short (distance between molecules in nanoscale). In the RIMR reaction, the enzyme is immobilized on the microchannel sidewalls, and the reactant in the solution need to diffuse to the sidewalls to react with the enzyme. The average diffusion distance is about $20\text{ }\mu\text{m}$ (half of the microchannel height), which is much larger than that of the bulk reaction. For better clarity, the sentence is modified in the revised manuscript.

In addition, the immobilized enzyme may lose the flexibility to bind the natural ligands during catalysis and the diffusion distance for the reactant to the immobilized enzyme on the microchannel sidewalls becomes relatively long (as compared to the free enzyme)⁴².

--- Page 7, paragraph 1, lines 122 – 125

Comment 4: Lines 172-174 and Fig 6b: Although HPLC measurements require a calibration curve to be used quantitatively, the area of the RuBP peak from RIMRs should appear clearly

lower, compared to that obtained from BIMRs, as in this case the reaction led to the formation of 3-PGA. Why the RuBP peaks from RIMRs and BIMRs are identical?

Response 4: Thanks to the Reviewer for having pointed out this problem. On the revision stage, we have conducted new HPLC-MS/MS analyses of the solutions obtained by RIMRs and BIMRs and have updated the results to Fig. 2b in the revised manuscript.

During our experiment, we found that the buffer we use contains lots of salt ions, which would have a great impact on the signal of RuBP and 3-PGA. Large carry-over of signals occurs and the calibration between the peak areas and the concentration is hard to achieve. To obtain better result, we use a shorter column and reoptimize the conditions of HPLC-MS/MS. All the standard samples are dissolved in the mixture of buffer and ACN (50:50, v/v) to reduce the influence of salt ions in buffer. For the determination of the produced 3-PGA by RIMRs, the production solutions are diluted by the same volume of ACN before the analysis. The new result is updated in Fig. 2b, which shows a significant difference in the areas of RuBP peaks from RIMRs and BIMRs. On such a condition, the calibration curves of the standard RuBP and 3-PGA with respect to the peak areas are obtained (see Fig. S10).

Fig. 2 b HPLC-MS/MS chromatography of RuBP and 3-PGA in the production solutions I obtained from RIMRs and BIMRs at the reaction time of 5 min.

Fig. S10 (a) HPLC-MS/MS chromatography of the standard RuBP and 3-PGA with the concentration of 100 μM . (b) Calibration of the peak areas of the RuBP and 3-PGA chromatography as a function of their concentrations.

However, no significant 3-PGA amount can be detected from the production solution I of the BIMRs. It is also noted that the RuBP signal from BIMRs is higher than that from RIMRs, indicating the conversion of RuBP to 3-PGA in RIMRs.

--- Page 8, paragraph 2, lines 159 – 161

Comment 5: Lines 190-197 and Fig 7b: The authors incubated the samples in oven for 10 mins (line 368) in order to test their thermal stability, but I think that this kind of experiment is not informative of the real stability of the RIMRs. In my opinion, two different test should be made to evaluate the possibility to use the RIMRs in 3-PGA production: the first with incubation time coherent with the retention time of the substrate in the microreactor (depending on the chosen flux), and the second with a prolonged use of the RIMRs for the continuous production of 3-PGA.

Response 5: In our stability experiments, the reaction time is 1 min. The incubation of 1 min causes very small reduction of the enzyme activity, which is within the error range ($\sim 10\%$) and difficult to measure. The 10-min incubation time is already 10 folds of the reaction time which can well represent the thermal stability for the one-cycle reaction. Following the Reviewer's suggestion, we have conducted more incubation experiments for other durations (0 min to 60

min) at 50 °C, as shown in Fig. S12. After the incubation for 60 min, the immobilized RuBisCO retains 76% of its initial activity while the free RuBisCO only retains 43% of its initial activity.

Fig. S12 Relative activities retained after different incubation time at 50 °C for the free RuBisCO and the immobilized RuBisCO. The amount of RuBisCO used is 21.875 µg for both the immobilized and free ones. The collected production solutions are 100 µL.

Comment 6: Line 246: I do not agree with authors' affirmation about the “excellent reusability” of RIMRs. Their data about RuBisCO activity after the immobilization on the micro reactors are good, but a 10% decrease of enzymatic activity after only 5 reaction cycle is not sufficient to hypothesize a real use of RIMRs. More data are needed in order to understand if the slightly negative trend shown in Figure 7c could accelerate or slow down after other reaction cycles.

Response 6: Thanks for the suggestion. We have changed the term to “good reusability” in the revised manuscript. Although the “excellent reusability” of our RIMRs cannot be affirmed, the reusability of the RIMRs is still better than that of the free RuBisCO which is usually hard to recycle.

To find the trend of reusability, we have conducted new experiments to test the RIMRs for up to 10 reaction cycles. As one can see from Fig. 3c, the relative activity decreases with the increase of cycles but the trend of drop gradually slows down. The activity retains 90.4% after 5 cycles and 78.5% after 10 cycles. The reusability at different flow rates from 7 to 0.7 µL·min⁻¹ is also conducted (see Fig. 3d and Fig. S13) to check if a slower flow rate would help alleviate the activity loss. Fig. 3d shows that the flow rate of 1.4 µL·min⁻¹ yields the highest remaining activity after 10 reaction cycles. The reasons are presented in the revised manuscript as follows.

It also exhibited good reusability with 90.4% activity retained after 5 cycles of reuse and 78.5% after 10 cycles.

The RIMRs also presented a good reusability with 90.4% of its initial activity retained after 5 cycles of reuse and 78.5% after 10 cycles.

Fig. 3 c Reusability of the RIMRs when RuBP is injected at $1.4 \mu\text{L}\cdot\text{min}^{-1}$ (the reaction time is 5 min). Red dash-dotted line is the third-order polynomial fitting, representing that the activity drop trend tends to slow down with the increase of cycles of reuse. **d** Relative activity as a function of the flow rate of RuBP injection (from $7 \mu\text{L}\cdot\text{min}^{-1}$ to $0.7 \mu\text{L}\cdot\text{min}^{-1}$) for repeated uses.

As shown by the grey bars in Fig. 3c, the relative activity decreases with the increase of cycles of reuse but the trend of activity drop gradually slows down (red dash-dotted line by third-order polynomial fitting). More specifically, 90.4% and 78.5% of initial activity can be retained after 5 cycles and 10 cycles of reuse, respectively, when the flow rate is kept at $1.4 \mu\text{L}\cdot\text{min}^{-1}$. The activity loss after long-time assay may be attributed to two factors. The first factor is the enzyme inactivation after repeated uses and the inhibition of accumulative 3-PGA and RuBP degradation products in aqueous solution⁴⁵. Therefore, RuBisCO activity is usually measured in a short assay that is stopped in less than 1 min⁴⁸. The second factor is the enzyme detachment due to the flushing by the running flow. When the flow rate is lower down, the flow in the microchannel moves slower but the reaction time and the operation time is extended. Correspondingly, the flushing effect is reduced but the inactivation effect is increased, and their combination affects the reusability of RuBisCO in RIMRs. Based on the measured reusability at varying flow rates at 30°C (see Fig. 3d and Fig. S13), the flow rate of $1.4 \mu\text{L}\cdot\text{min}^{-1}$ yields the highest remaining activity after 10 cycles of reuse. The reusability of RuBisCO in the bulk reaction is hard to determine since it needs special techniques to filter out and recycle the enzyme. In contrast, it is quite facile in the RIMRs reaction by pumping out the production solution and then injecting the new reactant mixture.

The reusability of RIMRs was evaluated by injecting the reactant mixture into the reactor for ten cycles of reuse at different flow rates. The reusability of RuBisCO at different flow rates are plotted in Fig. S13a. The relative activities after different cycles of reuse are plotted as a function of flow rate in Fig. S13b.

Fig. S13 Reusability of the RIMRs for different cycles of reuse when the reactant mixture is injected at the flow rates of $7 \mu\text{L}\cdot\text{min}^{-1}$, $1.4 \mu\text{L}\cdot\text{min}^{-1}$, $1 \mu\text{L}\cdot\text{min}^{-1}$ and $0.7 \mu\text{L}\cdot\text{min}^{-1}$. As the volume of reactor is $7 \mu\text{L}$, the corresponding reaction time is 1 min, 5 min, 7 min and 10 min, respectively. (a) Reusability as a function of cycle of reuse at different flow rates. (b) Reusability as a function of flow rate after different cycles of reuse.

--- Page S14, session 10 of Supplementary Information

Comment 7: Lines 258-259: The sentence “A promising industrial route to produce large amount of basic food material” it does not seem appropriate to summarize these results if we consider the dimensions of the microreactor respect to the amount of food requested to resolve “the food crisis relief and future space colonization”.

Response 7: We agree with the comment that it is a bit too far for this work to jump directly to real industrial production. In the revised manuscript, we have softened the statement and have also added some linkage in the revised manuscript.

This work, though still far from industrial production, makes the first move towards replicating the whole light-independent reactions of NPS to artificially produce basic food materials, which may hold the key to food crisis relief and future space colonization.

--- Abstract, the last 4 lines

Moreover, the immobilization amount of RuBisCO is still small due to the limited space of the microreactor. Some new material such as MOFs⁵⁰, COFs⁵¹, mesoporous nanoparticles⁵², and carbon nitride⁵³ could be applied to the microreactor to provide larger surface areas in the future. The reactor can also be easily scaled up⁵⁴⁻⁵⁷ to increase the output and more functionalities can be integrated, such as deoxygenation, temperature control of individual reaction, etc.

--- Page 15, paragraph 2, lines 274 – 278

In a long run, a scientific approach would be available to produce massive basic food materials to relieve the food crisis and to prepare for future space colonization.

--- Page 15, paragraph 2, the last 2 lines

Comment 8: Lines 324-325: How was the surface area of the microreactor determined?

Response 8: The surface area is determined from the estimated volume of the microreactor (V_r), the width (w_c) and the height (h_c) of the microchannel. In brief, the serpentine microchannel is treated as a long folded microchannel with a rectangular cross section. The total volume of the microreactor is estimated by calculating the volume of the infused water, which is about 7 μL . The width and the height of the cross section are 800 μm and 40 μm , respectively. The length of the folded microchannel (l_c) should be $l_c = \frac{V_r}{w_c \cdot h_c} = 218.75 \text{ mm}$. Then the surface area is the total areas of the sidewalls of microchannel as given by: $A_r = 2 \times (w_c + h_c) l_c = 367.5 \text{ cm}^2$.

Comment 9: All the text: Some errors are present. Please edit the manuscript and check it for typo errors.

Response 9: Following the Reviewer's suggestion, we have carefully and thoroughly checked the manuscript to correct the typos.

Response to Reviewer 3

General comment 1: This manuscript describes the immobilization of a protein onto a solid surface by amine coupling followed by characterization of its activity and various properties such as thermal stability. This is a very routine type of experiment and thus is not well suited for a non-specialty journal. I believe there are engineering journals that will publish this type of analysis.

General response 1: We appreciate the critical comments by the Reviewer, but still hope to convince him of the originality and significance of this work. The originality of this work lies mostly in the first try of replicating the first part of Calvin cycle by immobilizing RuBisCO in microreactors. Regarding the scientific significance, this work, though still far from industrial production, makes the first move towards replicating the whole light-independent reactions of NPS to artificially produce basic food materials, which may hold the key to food crisis relief and future space colonization. Below we will present more detailed explanations.

Indeed, enzyme immobilization has been well studied over the recent decades. However, the study of RuBisCO immobilization is still elusive (*Biotechnol. Bioeng.* **81**, 705-711, (2003); *Smart Mater. Struct.* **25**, 125033, (2016); *Biotechnol. Biofuels* **10**, 175, (2017)). The major challenge of RuBisCO immobilization is the unavoidable activity reduction when immobilization by amine coupling. In addition, the immobilization by the traditional EDC-NHS coupling is hard to achieve due to the exist of the primary amino groups in Tris-HCl buffer. And the NHS activity would drop rapidly due to hydrolysis of the NHS ester under alkaline conditions (*Adv. Mater.* **21**, 431-434, (2009)). Here, we apply polydopamine as the linker to immobilize RuBisCO on PDMS microreactors, which is quite simple and effective. And no previous research is found to use polydopamine to immobilize RuBisCO.

Moreover, the RuBisCO immobilization is achieved, for the first time, in a microfluidic reactor instead of being immobilized by a matrix or a carrier. The use of microfluidics brings in many advantages (e.g., continuous production using a small amount of enzyme (Fig. 4), easy control of reaction condition (Fig. 2a), avoidance of feedback inhibition (Fig. 2a & Fig. S9), easy collection of the products, etc.) and makes it feasible to streamline different enzymatic steps for a complete Calvin cycle. The research is a multi-discipline work including physics, chemistry, and biology. There are many high impact studies on the combination of enzyme immobilization and microfluidics (*Commun. Biol.* **2**, 42, (2019); *Sci. Adv.* **4**, eaat2816, (2018); *Angew. Chem. Int. Ed.* **57**, 17028-17032, (2018); *Nat. Nanotechnol.* **11**, 409, (2016); *Nat. Commun.* **8**, 663, (2017); *Small* **8**, 3531-3537,

(2012); *J. Am. Chem. Soc.* **133**, 6006-6011, (2011)) and the advantages are not negligible.

Furthermore, the major aim of the research is to produce the glucose precursor continuously, but not to simply characterize the properties of RuBisCO after immobilization. 3-PGA can be produced constantly (Fig. 4). This is ascribed to the continuously pumping out of the products, therefore avoiding the feedback inhibition. Also, in the microenvironment of the microfluidic reactor, the concentration of RuBisCO keeps the same regardless of the increasing volume of RuBP. However, for the bulk reaction, the increasing volume of RuBP leads to a decrease of RuBisCO concentration, causing a significant reduction of the reaction rate at large volume.

Fig. 2 Feasibility of RIMRs. **a** Production amount of 3-PGA as a function of the reaction time for the RIMRs (solid red circles) and the BIMRs (open dark squares). The red dash-dotted line and dark dashed line are the liner fitting regressions. The slope of liner fitting regression of 3-PGA production in RIMRs is $13.8 \mu\text{mol}\cdot\text{g}^{-1} \text{RuBisCO}\cdot\text{min}^{-1}$ and that in BIMRs is almost 0. The initially injected RuBisCO and BSA concentrations for immobilization are both $6.25 \mu\text{g}\cdot\mu\text{L}^{-1}$. The collected production solutions I from RIMRs and BIMRs are 21 μL . Reaction temperature is 30 °C.

Fig. S9 Production of 3-PGA by free RuBisCO as a function of time. The amount of RuBisCO used is 21.875 µg. Production solution II is collected for 21 µL.

In contrast, the RIMRs demonstrate a linearly increased production of 3-PGA at the rate of $13.8 \mu\text{mol}\cdot\text{g}^{-1} \text{RuBisCO}\cdot\text{min}^{-1}$ (red solid circles and red dash-dotted fitting line of Fig. 2a) when the reaction time is 5 min or shorter. For free RuBisCO in the bulk reaction, things become very different, the 3-PGA production at the reaction time of 5 min is close to that at 2 min. This may be attributed to the inhibition of RuBP degradation products in aqueous solution and the feedback inhibition of the products 3-PGA⁴⁵. In the RIMRs reaction, the products are flowed through the reactor and pumped out immediately after reaction. The accumulation of 3-PGA around RuBisCO is reduced and the feedback inhibition is therefore alleviated. This is one of the core features of RIMRs.

--- Page 8, paragraph 1, lines 141 – 149

Fig. 4 Continuous production of 3-PGA as a function of the volume of used reactant mixture solution, the dark open squares represent the 3-PGA production in the bulk reaction with free RuBisCO and the red solid circles represent the 3-PGA production in the RIMRs with the immobilized RuBisCO. The amount of RuBisCO used is 21.875 µg for both the immobilized and free ones. Reaction temperature is 30 °C. Reaction time is 1 min.

Nevertheless, the 3-PGA production by the RIMRs in the continuous mode (reaction temperature: 30 °C, reaction time: 1 min) increases with V_p at a rate of $0.39 \mu\text{mol}\cdot\text{g}^{-1} \text{RuBisCO}\cdot\text{min}^{-1}\cdot\text{L}^{-1}$ (red solid circles in Fig. 4). In contrast, the 3-PGA production in the bulk reaction goes up gradually and tends to saturate at about $360 \mu\text{mol}\cdot\text{g}^{-1} \text{RuBisCO}$ when V_p is larger than 600 μL (dark open squares in Fig. 4). The saturation of 3-PGA production in the bulk reaction is attributed to the reduced RuBisCO concentration with the increased V_p when the amount of RuBisCO remains the same. Regarding the RIMRs reaction, the increase of V_p does not affect the RuBisCO concentration thanks to the immediate pumping out of the production solution I and the continuous injection of fresh reactant mixture, which also help to avoid the feedback inhibition. Together with the good reusability of immobilized RuBisCO, the RIMRs enable the accumulative production of 3-PGA using only a small amount of RuBisCO. This is another core merit of RIMRs.

--- Page 13, paragraph 2, lines 239 – 249

General comment 2: The authors purport to be testing some type of artificial photosynthesis, but I don't think this is an accurate description. If you add RuBP and CO₂ to Rubisco you will get 3PG and this is not a recent discovery. In the same vein you could perform any other reaction in the Calvin cycle and claim it to be APS. In fact, the Calvin cycle is independent of light, often being powered chemically, it would more accurately be described as artificial chemosynthesis.

Response 2: We agree with the Reviewer on this issue. We understand that this 3-PGA production reaction in Calvin cycle is independent of light and needs CO₂ and RuBP. Therefore, we only stated “artificial synthesis” before. It would be more accurate if more information is added into the title. But in order to keep the title concise and informative, we make some modification to the tile as “Continuous artificial synthesis of glucose precursor using enzyme-immobilized microfluidic reactors”. We also make some other modifications in the revised manuscript as below.

This work presents a scientific solution to overcome this problem by immobilizing RuBisCO into a microfluidic reactor, which demonstrated a continuous production of glucose precursor at $13.8 \mu\text{mol}\cdot\text{g}^{-1} \text{RuBisCO}\cdot\text{min}^{-1}$ from CO₂ and ribulose-1,5-bisphosphate.

--- Abstract, lines 7 – 9

The reaction performance of artificial chemosynthesis of glucose precursor in the RuBisCO-immobilized microfluidic reactors (RIMRs) is compared with that in the bulk reaction in terms of the stability, reusability and long-term 3-PGA production subsequently.

--- Page 5, paragraph 1, lines 82 – 85

To check the feasibility of the reactors for artificial chemosynthesis of glucose precursor, the prepared RIMRs are used to produce 3-PGA from RuBP and CO₂ under different reaction times.

--- Page 7, paragraph 2, lines 136 – 137

It can also continuously produce 3-PGA by constantly injecting CO₂ and RuBP, enabling the accumulative production of 3-PGA with very small amount of RuBisCO in the continuous mode.

--- Page 14, paragraph 1, lines 265 – 267

General comment 3: A key problem is already seen in the title: “continuous artificial synthesis of glucose precursor from CO₂”. This is wrong – shown is synthesis of glucose precursor from CO₂ and ribulose 1,5-bisphosphate- a high energy compound that is purchased at the cost of ~\$1000/ 100 mg from Sigma, which indicates that feeding the world using this concept would be quite challenging.

Response 3: We have also noted the high cost of RuBP. Nevertheless, RuBP can be regenerated by the enzymatic reactions (i.e., Phase 3 of Calvin cycle) *in vivo*. In the future, the RuBP regeneration can be integrated into the microchip in order to complete the whole Calvin cycle reactions for continuous glucose production in a self-sustained matter.

General comment 4: To claim APS it is necessary to have light and then obtain some type of energy storage chemical. I don't see how it is sufficient to demonstrate that one is able to replicate an assay of the world's best studied enzyme.

I could envisage work in this vein to become interesting for Nature Communications if the current title was in fact accurate – conversion of CO₂ to G3P- not assaying a single exceedingly well studied enzymatic reaction.

Response 4: This work focused on “artificial synthesis” as stated in the manuscript title. It is a part of the artificial photosynthesis (APS).

Regarding RuBisCO, it has been extensively studied for decades and regarded as a barrier to increasing photosynthetic rate due to its low catalytic rate and the poor specificity. Scientists have paid lots of attention on RuBisCO, especially in the genetic engineering, to improve the photosynthesis (*Nature* **566**, 131-135, (2019); *Science* **358**, 1272-1278, (2017); *Nat. Commun.* **6**, 8883, (2015); *Nature* **513**, 547, (2014); *Proc. Natl. Acad. Sci. U.S.A.* **109**, 478-483, (2012); *Crit. Rev. Biotechnol.* **32**, 1-21, (2012)). But they are mainly designed for *in vivo* biological systems (*Plant Cell Environ.* **38**, 1817-1832, (2015); *Acta Physiol. Plant.* **36**, 3101-3114, (2014)). It is still challenging, time-consuming and labor intensive to balance the metabolic pathways in intact cells. We are trying to move towards another direction to improve the RuBisCO properties by the immobilization in microreactors. Although there are some problems in current stage, the efforts may inspire the researchers to new discoveries and technical advances. We have also mentioned the reason we choose to immobilize RuBisCO in the manuscript.

The production of 3-PGA to G3P can be achieved with another two enzymes PGK and GAPDH, which are already used in our activity assay. This cascaded reaction by three enzymes are our new research target. This work is just a start point and lays the foundation for future research of glucose synthesis.

The motivation of our work has been explained in the manuscript as follows:

Although many studies have been conducted to improve the efficiency of RuBisCO¹⁵⁻²⁰, they are mainly designed for *in vivo* biological systems^{21,22}. Lots of energy and labor are consumed to balance the metabolic pathways in intact cells. For the industrial cell-free application with the demands of sustainability and scalability, immobilizing and concentrating RuBisCO into man-made structures would be a more promising method to enhance the overall catalytic efficiency²³⁻²⁵.

--- Page 4, paragraph 1, lines 53 – 58

Comment 1: Regarding Figs. 2-5 and Results p. 5-7

A large portion of this manuscript describes the immobilization of a protein on a solid surface. This is a trivial procedure, routinely performed in biophysics in applications such as surface plasmon resonance and does not warrant significant characterization to be described in the main text of a high impact publication. Such

characterization should however be carefully described in the supplementary information and occupy at most 2-3 sentences in the main manuscript. However, the presented characterization is only meaningful if it is also performed for a control- hence the same should be done for the BSA immobilization (that presumably will result in identical data?)

Response 1: Following the Reviewer's suggestion, we have moved the data of fluorescence images, water contact angles and some SEMs to the Supplementary Information. The corresponding descriptions are also refined in the main text and more details of them are provided in session 2 of revised Supplementary Information. The three-dimensional diagram is kept in the main text to illustrate the working principle of our microreactor (new Fig. 1b). Only one of SEM images (new Fig. 1c) is kept in the main text to show the surface changes after RuBisCO immobilization, in which RuBisCO is distinguished from PDA layer by color. The protein loading results are needed to optimize the RuBisCO concentration, which is an important parameter for the following experiments, therefore they are kept in the main text (new Fig. 1d). All the characterizations are also performed for the BSA-immobilized microreactor as the control experiments (new Fig. S2 – S6). The results are very similar with the RuBisCO-immobilized microreactor.

Fig. 1 c SEM image of the inner surfaces of RIMRs. Flat and smooth PDMS becomes rough and is covered by PDA nanoparticles after PDA modification (brownish color). Large blocks are the immobilized RuBisCO (green blocks).

The three-dimensional diagram and the photograph (inset) of the RIMRs is presented in Fig. 1b. It consists of two PDMS layers, and the inner surfaces of microchannels are functionalized by PDA. The abundant catechol groups of PDA layer can react with the amine groups of RuBisCO through the Michael addition or Schiff base reactions³⁸, facilitating the immobilization of RuBisCO (Fig.

S1). Fig. 1c is the SEM image of the microchannels' inner surfaces after the RuBisCO immobilization. The PDA modified PDMS is coarse and rough and is covered with many PDA nanoparticles. The RuBisCO immobilization then leads to the formation of some blocks, proving the successful immobilization (see also Fig. S3). The successful immobilization of RuBisCO is further confirmed by the measurements of water contact angle, Raman and ATR-FTIR spectrum and fluorescence images of the microfluidic channels (see Figs. S4 – S6).

--- Page 5, paragraph 2, lines 91 – 100

Fig. S2 Optical images of microchannel. (a) Pristine PDMS microchannel, (b) and (d) PDA modified microchannel, (c) RuBisCO-immobilized microchannel, (e) BSA-immobilized microchannel. The scale bar is 500 μm .

Fig. S2 is the optical microscopic images obtained from Olympus BX41 of the microchannels after each step of the immobilization procedures. After the PDA modification, the transparent PDMS channels become opaque and brown. While the subsequent RuBisCO immobilization or BSA immobilization would not change the surface roughness.

--- Page S2, session 2.1 of Supplementary Information

Fig. S3 SEM images. (a) pristine PDMS surface, (b) and (c) PDA-modified PDMS surface, (d) RuBisCO-immobilized PDA-PDMS surface at the magnification of $10000\times$. (e) and (f) BSA-immobilized PDA-PDMS surface.

Fig. S3 is the supplementary SEM images obtained from JEOL JSM-6490 for the surface characterization of the microchannels' inner surfaces. The pristine PDMS surface is flat and smooth (Fig. S3a). While after the PDA modification, a rough layer with many nanoparticles is formed (Fig. S3b and S3c). This coarse PDA layer can provide a much larger surface area than the smooth PDMS surface, offering more active functional groups to covalently couple with RuBisCO. Then, the immobilization of RuBisCO brings about many blocks onto the PDA layer (Fig. S3d). The blocks formation can also be observed when BSA is immobilized into the microreactors (Fig. S3e and S3f).

--- Page S3, session 2.2 of Supplementary Information

Fig. S4 Water contact angles of PDMS, PDA-modified PDMS, RIMRs and BIMRs surfaces. All data are obtained for three times. Source data are provided as a Source Data file.

Fig. S4 shows the observed water contact angles conducted by a standard contact angle goniometer (Model 200, ramé-hart instrument co.). The water contact angle of the inner surface significantly decreases from 103.8° to 25.5° after the PDA modification. This hydrophilicity improvement results from the abundant catechol groups of PDA. In contrast, further immobilization of RuBisCO leads to an increase of water contact angle, which is probably due to the hydrophobic side chains of amino acids in RuBisCO. For the control experiments, the BSA immobilization also presents similar results.

--- Page S4, session 2.3 of Supplementary Information

Fig. S5 Raman spectra (a) and ATR-IR spectra (b) of the pristine PDMS (black line), RuBisCO_PDA_PDMS (red line) and BSA_PDA_PDMS (blue line). Source data are provided as a Source Data file.

A Raman spectrometer (Witec_Confocal Raman system) and an attenuated total reflection Fourier transform infrared spectroscopy (ATR-FTIR, BRUKER) are used to obtain the corresponding spectrum for the confirmation RuBisCO immobilization. As shown in Fig. S5, new peaks (1368 cm^{-1} and 1564 cm^{-1} in Raman spectra and $1400\text{-}1800\text{ cm}^{-1}$ and $3000\text{-}3750\text{ cm}^{-1}$ in ATR-FTIR spectra) can be observed after the PDA modification and the RuBisCO/BSA immobilization when compared with the pristine PDMS microfluidic channels. Particularly, the FTIR spectrum of the RIMRs has a weak band at about 1600 cm^{-1} , which could be referred to the vibrational modes of the peptide bond (amide bands). This absorption band confirms the effective RuBisCO immobilization and gives information about conformational changes induced by the immobilization procedure.

--- Page S5, session 2.4 of Supplementary Information

Fig. S6 Fluorescence experiments for confirming RuBisCO immobilization. (a) Fluorescence images of the empty microchannel, the RuBisCO-FITC filled microchannel, and the rinsed microchannel. (b) The corresponding fluorescence intensity profiles obtained along the observation lines. (c) Fluorescence images and (d) fluorescence intensity profiles of the BSA-FITC filled microchannel and the rinsed microchannel. The scale bar is 500 μm .

The fluorescence experiment was also conducted to verify the RuBisCO immobilization. The RuBisCO-FITC (RuBisCO tagged by fluorescein isothiocyanate) solution was introduced into the PDA-PDMS microfluidic reactor and then kept at room temperature for 6 hours. The reactor was observed under a fluorescence microscope (Olympus BX41) to record the fluorescence images. Afterwards, it was thoroughly rinsed by 0.1-M phosphate-buffered saline (PBS, pH 9.2) to check whether the fluorescence emission still existed. As shown by the fluorescence images in Fig. S6a and the corresponding fluorescence intensity profiles in Fig. S6b, the fluorescence intensity difference (noted as $\Delta(FI)$) between the microchannel and the microchannel wall is significantly increased after the injection of the RuBisCO-FITC solution. After the thoroughly rinsing by the PBS buffer, $\Delta(FI)$ decreases but not drops to zero. The non-zero $\Delta(FI)$ after the rinse proves that a portion of RuBisCO is well retained on the microchannel surface and the strategy of RuBisCO immobilization on microfluidic reactor via PDA modification is feasible. Similar results are also observed in the BSA-immobilized microfluidic reactors as shown in Fig. S6c and d.

--- Page S6, session 2.5 of Supplementary Information

Comment 2: Table 1: the reported V_{max} when converted to turnover per active site for the bulk reaction is equivalent to $\sim 3 \text{ min}^{-1}$ – however Spinach rubisco has a k_{cat} of about 2 s^{-1} , this suggests the enzyme that is being used here (powder purchased

from sigma) is essentially non-functional. maybe there was a calculation error somewhere? Alternatively the unusual (relying on signal amplification rather than a direct readout) spectrophotometric assay used here may not have been implemented successfully. This needs to be carefully checked.

One issue to check carefully concerns the inclusion of significant amounts of ethanol (used to stop the reaction) in the coupled assay- maybe this is responsible for the very low activities reported. It would help to present the original data used to derive the reported activities in the supplementary information (not just the 3PG calibration curve).

Related to this the activity reported in Fig. 6a is also incredibly slow (inactive) with one active site producing 0.5 molecules of 3PG per minute.

Response 2: Thanks to the Reviewer for having pointed out this problem. When preparing for the revision, we have conducted more experiments to carefully determine the kinetic parameters of the immobilized RuBisCO and free RuBisCO by reconducting the calibration. The results are updated to Table 1. From the results, similar conclusion can still be obtained. After immobilization, the value of K_m (RuBP) becomes larger and V_{max} becomes smaller. The corresponding k_{cat} of free RuBisCO is about 0.19 s^{-1} , which is indeed smaller than the natural k_{cat} of RuBisCO from Spinach. The major reason may be due to that RuBisCO from Sigma (Catalogue # R8000) is partially purified. The activity given by Sigma is only 0.01~0.1 unit/mg, where one unit can transfer one molecule RuBP and CO_2 to two molecule 3-PGA. In our experiments, the obtained maximum activity would be ~ 0.08 unit/mg, which is within the range of the reference data given by Sigma.

Table 1: Kinetic parameters obtained from RIMRs reaction and bulk reaction.[&]

	V_{max} ($\text{mmol} \cdot \text{min}^{-1} \cdot \text{g}^{-1}$ RuBisCO)	K_m (RuBP) (mM)
RIMRs reaction	0.069 ± 0.003	0.067 ± 0.014
Bulk reaction	0.169 ± 0.003	0.048 ± 0.004

[&] The collected production solutions I and II are both 100 μL .

In general, four representative methods are used to determine RuBisCO activity, the radioactive assay based on incorporation of $^{14}\text{CO}_2$ into 3-PGA (*Anal.*

Biochem. **78**, 66-75, (1977); *Proc. Natl. Acad. Sci. U.S.A.* **82**, 8024-8028, (1985)), the spectrophotometric assay based on the coupling of several enzymes to read out the decrease of NADH (*Biochim. Biophys. Acta, Enzymology* **358**, 226-229, (1974)), the chromatography assay based on 3-PGA amount determination by separation (*Plant Cell Physiol.* **37**, 325-331, (1996); *J. Biochem. Bioph. Methods* **52**, 179-187, (2002)) and the amplification signal assay (the stopped assay) used here (*Plant Cell Environ.* **30**, 1163-1175, (2007); *Plant Methods* **10**, 17, (2014)). The most widely used radioactive assay is usually affected by constraints on the use of radioactivity. The non-radioactive assays are usually not sensitive enough, especially when our RuBisCO activity is relatively low. Therefore, the amplification assay is used here to amplify the signal of NADH decreasing. This method is relatively sensitive, cheap and requires only standard laboratory equipment (*Plant Cell Environ.* **30**, 1163-1175, (2007)). As reported by the paper using the amplification assay, the determined values of K_m (RuBP) and V_{max} of RuBisCO from raw extracts of different leaves are similar to the values reported in the BRENDA database (<https://www.brenda-enzymes.org/>).

In our test using the amplification assay, the same volume of 80% ethanol is used to stop the 3-PGA production reaction and then 20 μ L of the mixture is added into the solution with a total volume of 100 μ L for the NADH decreasing analysis. The final content of ethanol is only 8%. To determine the 3-PGA amount more accurately, we have carefully calibrated 3-PGA in both the buffer and the buffer:ethanol (60:40, v/v) mixture (Fig. S8b). The original data of the absorbance decreasing obtained with different concentrations of 3-PGA dissolved in buffer are also plotted in Fig. S8a. All the data of the immobilized RuBisCO and the free RuBisCO are recalculated to determine the 3-PGA amount according to the 3-PGA calibration lines in the buffer and the buffer:ethanol mixture, respectively.

Fig. S8 (a) Decrease of the absorbance at 340 nm as a function of the time for different concentrations (from 0.0025 to 0.05 M) of 3-PGA dissolved in the reaction buffer. (b) Calibration curves of 3-PGA both in the reaction buffer and in the mixture of reaction buffer:ethanol (60:40, v/v) mixture by the amplification signal assay with using a UV-Vis spectrometer.

Although the activity of RuBisCO used here is not high, we cannot simply conclude that RuBisCO is inactive for 3-PGA production. From the HPLC-MS/MS chromatography, the signal of 3-PGA can be well detected from the production solution collected from the RIMRs with the 5-min reaction time while there is no signal from the BIMRs (see new Fig. 2b). We have also conducted the calibration of 3-PGA from the obtained HPLC-MS/MS chromatography of standards (new Fig. S10). The derived amount of 3-PGA is about $71.9 \mu\text{mol}\cdot\text{g}^{-1}$ RuBisCO from the peak area of HPLC-MS/MS chromatography, which is close to the produced 3-PGA amount of $68.2 \pm 5.4 \mu\text{mol}\cdot\text{g}^{-1}$ RuBisCO in 5 min as determined from the UV-Vis spectrum. Therefore, the amplification assay used here is reliable for the determination of 3-PGA amount.

Fig. 2 b HPLC-MS/MS chromatography of RuBP and 3-PGA in the production solutions I obtained from RIMRs and BIMRs at the reaction time of 5 min.

Fig. S10 (a) HPLC-MS/MS chromatography of the standard RuBP and 3-PGA with the concentration of 100 μM . **(b)** Calibration of the peak areas of the RuBP and 3-PGA chromatography as a function of their concentrations.

Comment 3: Pg. 2 “we also carried out a series of studies on the light-dependent reactions... very high efficiency“ – this is a strange claim in the absence of a reference. Either remove or provide citation.

Response 3: The absence of the reference is due to the requirement of double-blind review process. We cannot cite the papers from our group in the manuscript. The citation will be added before the final publication.

Comment 4: Fig. 1b- the chemical structures of amine linked Rubisco to the surface does not look correct to me (Nitrogen bonding covalently to the center of an aromatic ring)- please check

Response 4: The structures have been corrected in Fig. S1 referring to the paper (*ACS Appl. Mater. Interfaces* **5**, 10559-10564, (2013)). The binding of amine to the center of the aromatic ring I used to illustrate means that the amine may be added to the ortho-position or the para-position of hydroxyl on the aromatic ring by the Michael addition (*Nanoscale* **3**, 4916-4928, (2011); *Adv. Mater.* **21**, 431-434, (2009); *Nat. Med.* **11**, 1214, (2005).). Here, the illustration of amine binding to the aromatic ring at the para-position of hydroxyl is taken as an example.

Fig. S1 Illustration of the procedures of RuBisCO immobilization.

In Fig. S1c, the amine binding to the aromatic ring at the para-position of hydroxyl is a possible structure for the bonding of RuBisCO to PDA by Michael addition^{S1}.

--- Page S1, Paragraph 1, the last 3 lines

Comment 5: The legend to Fig. 7a does not indicate the temperature used in this experiment. (the methods says 4 deg C – you also need to define concentration and buffer conditions)- the loss of activity is in fact concerning, but should not be observed. Rubisco is a very stable enzyme. This should be characterized more carefully- does migration behaviour on a native gel change over time? Is degradation observed on SDS-PAGE? Does the protein precipitate? The loss of activity may suggest the enzyme preparation (from Sigma) is still quite impure.

Response 5: Following the Reviewer's suggestion, we have added the reaction conditions to all figures. Moreover, we have carried out the SDS-PAGE tests to analyze the free RuBisCO after different storage time, as shown by Fig. S11 in session 8 of Supplementary Information. Generally, RuBisCO is stable for several months if it is kept properly without Mg^{2+} and HCO_3^- at $-20\text{ }^\circ\text{C}$ (*Methods in Enzymology* Vol. 5 266-270 (Academic Press, Cambridge, Massachusetts, 1962)). Once Mg^{2+} and HCO_3^- are presented, significant decline of activity is observed (*Anal. Biochem.* **153**, 97-101, (1986); *Plant Physiology* **84**, 483-490, (1987)). Storage stability data is referred from values reported in the BRENDA database (<https://www.brenda-enzymes.org/>). Under our testing conditions (21.875 μg RuBisCO, 100 μL production solution II, reaction buffer: 100 mM Tris-HCl, 5 mM DTT, 66 mM KHCO_3 and 5 mM MgCl_2 , temperature $4\text{ }^\circ\text{C}$), the reduction of activity is obvious.

For the storage stability test of free RuBisCO, it was incubated in reaction buffer at $4\text{ }^\circ\text{C}$ for different days. No protein participants were observed after long time storage. However, the solution color changed from light yellow to brownish yellow (Fig. S11a).

RuBisCO with different incubation time were also prepared for the SDS-PAGE analysis, 40 μL of $5\text{-}\mu\text{g}\cdot\mu\text{L}^{-1}$ RuBisCO in the reaction buffer incubated for different days were added to 25- μL SDS dye and then boiled for 10 minutes. Next, 8 μL of the mixtures were loaded to SDS-PAGE (7% SDS gel). As shown in Fig. S11b, the most significant protein bands are detected at $\sim 54\text{ kDa}$ (A) and $\sim 13\text{ kDa}$ (C) which correspond to large subunits (LSU) and small subunits (SSU) of RuBisCO, respectively. The color of protein band of LSU tends to be lighter with the increase of incubation time, and that of RuBisCO incubated for 15 days is the lightest. Protein band appears at $\sim 40\text{ kDa}$ after days incubation, which may be the break-down product of LSU indicating the protein degradation after incubation^{S1,S2}. This explains that the protein degradation may be responsible for the activity drop of free RuBisCO after incubation.

Fig. S11 (a) Photograph of RuBisCO in reaction solution. (b) SDS-PAGE analysis of RuBisCO incubation in reaction buffer at 4 °C for different days. The most significant protein bands at ~ 54 kDa (A) and ~ 13 kDa (C) correspond to large subunits (LSU) and small subunits (SSU) of RuBisCO. Protein band at ~ 40 kDa (B) may be the break-down product of LSU. Each lane contains about 24.6 µg of RuBisCO. Staining was carried out with Coomassie Blue stain.

--- Page S12, session 8 of Supplementary Information

Comment 6: Related to above - define protein concentration and buffer conditions for every experiment- also the thermal stability assay. Fig. 7b does not explain the experiment in the legend.

Response 6: The reaction conditions are added to the corresponding legends. All the buffer conditions are the same as elaborated at the beginning of methods section (100 mM Tris-HCl, 5 mM DTT, 66 mM KHCO₃ and 5 mM MgCl₂).

Reviewers' comments:

Reviewer #1 (Remarks to the Author):

The manuscript was revised following the Reviewers' suggestions. I can recommend publication in Nature Communications

Reviewer #2 (Remarks to the Author):

The authors addressed all my concerns. I think the manuscript can be now accepted for publication.

Reviewer #3 (Remarks to the Author):

Review of "continuous artificial synthesis"

The authors have provided a revised manuscript that mostly addresses the various points made by the reviewers. As before I am not convinced that the work is in the league of Nature Communications. The following points need to be addressed to ensure technical soundness.

1-

"Furthermore, the major aim of the research is to produce the glucose precursor continuously, but not to simply characterize the properties of RuBisCO after immobilization. 3-PGA can be produced constantly (Fig. 4). This is ascribed to the continuously pumping out of the products, therefore avoiding the feedback inhibition."

I disagree with this interpretation as detailed:

Line 144-146 and Fig. S9- It is inconceivable that the result shown (new data) is due to feedback inhibition of 3-PGA (complete inactivity of Rubisco after two minutes). Feedback inhibition will result in a reduction in reaction rate, not abolishment. The only reasonable explanation here is that the substrate RuBP is exhausted (simple to test- add more RuBP and see whether more 3PG is made or not). Unfortunately I cannot calculate this directly, because the RuBP concentration used in the assay is not provided. The Figure legend needs to provide all substrate concentrations. Additional considerations are as follows: the ΔG for the carboxylation reaction is extremely negative, the equilibrium constant for this reaction is thus very high and in addition carboxylation is irreversible. In summary in a bulk reaction Rubisco will essentially use up all RuBP supplied, although the rate will decrease with time. The observed decrease in rate is also well characterized and due to the phenomenon known as fallover (not so much product inhibition). (Edmondson et al. Plant Physiology 1990- 93:1376-1397 – three papers). Fallover is due to tight binding of inhibitory sugar phosphates to the active site. A microfluidic system would not prevent this effect. Indeed the low Rubisco activities reported in this paper are likely due to a combination of fallover and irreversible inhibitor binding, since the RuBP used by the authors is well known to be highly contaminated by inhibitors (described by Ref. 45 in manuscript – Kane et al 1998 as well as Andralojc Biochem J. 2012 442:733-42)

2-

"From the results, similar conclusion can still be obtained. After immobilization, the value of K_m (RuBP) becomes larger and V_{max} becomes smaller. The corresponding k_{cat} of free RuBisCO is about 0.19 s^{-1} , which is indeed smaller than the natural k_{cat} of RuBisCO from Spinach. The major reason may be due to that RuBisCO from Sigma (Catalogue # R8000) is partially purified. The activity given by Sigma is only $0.01 \sim 0.1 \text{ unit/mg}$, where one unit can transfer one molecule RuBP and CO_2 to two molecule 3-PGA. In our experiments, the obtained maximum activity would be $\sim 0.08 \text{ unit/mg}$, which is within the range of the reference data given by Sigma"

You need to explicitly point out in your manuscript that compared to literature the activity of your Rubisco is very low. I suggest using a version of the explanations I have provided above regarding

fallover and the documented terrible quality of Sigma RuBP.

3-

“Comment 3: Pg. 2 “we also carried out a series of studies on the light-dependent reactions... very high efficiency” – this is a strange claim in the absence of a reference. Either remove or provide citation.

Response 3: The absence of the reference is due to the requirement of doubleblind review process. We cannot cite the papers from our group in the manuscript. The citation will be added before the final publication.”

This is a misunderstanding that will need editorial input. It is not acceptable to add references to a manuscript after peer review (since the accuracy of references need to be carefully evaluated by the reviewer), therefore in double blind the authors need to refer to such papers using neutral language (not “we showed”).

Other points:

Table 1: Explain what are the errors in this Table based on, why are they so small? Please also provide the true source data (time courses) that these numbers are based on in the Excel file, not just processed reaction rates.

Line 348: section 4, not session 4

Line 349: Define the substrate concentrations in the reactant mixture. This also applied to Line 102 and 105 of the supplementary information.

Response to Reviewer 3

General comment: The authors have provided a revised manuscript that mostly addresses the various points made by the reviewers. As before I am not convinced that the work is in the league of Nature Communications. The following points need to be addressed to ensure technical soundness.

Response: We feel sorry the previous version was still not good enough. Following the valuable suggestions of the Reviewer, we have conducted more experiments to strengthen the data and to clarify some unclear points. Hope it becomes satisfactory.

Comment 1: “Furthermore, the major aim of the research is to produce the glucose precursor continuously, but not to simply characterize the properties of RuBisCO after immobilization. 3-PGA can be produced constantly (Fig. 4). This is ascribed to the continuously pumping out of the products, therefore avoiding the feedback inhibition.”

I disagree with this interpretation as detailed:

Line 144-146 and Fig. S9- It is inconceivable that the result shown (new data) is due to feedback inhibition of 3-PGA (complete inactivity of Rubisco after two minutes). Feedback inhibition will result in a reduction in reaction rate, not abolishment. The only reasonable explanation here is that the substrate RuBP is exhausted (simple to test- add more RuBP and see whether more 3PG is made or not). Unfortunately, I cannot calculate this directly, because the RuBP concentration used in the assay is not provided. The Figure legend needs to provide all substrate concentrations.

Additional considerations are as follows: the ΔG for the carboxylation reaction is extremely negative, the equilibrium constant for this reaction is thus very high and in addition carboxylation is irreversible. In summary in a bulk reaction Rubisco will essentially use up all RuBP supplied, although the rate will decrease with time. The observed decrease in rate is also well characterized and due to the phenomenon known as fallover (not so much product inhibition). (Edmondson et al. Plant Physiology 1990- 93:1376-1397 – three papers). Fallover is due to tight binding of inhibitory sugar phosphates to the active site. A microfluidic system would not prevent this effect.

Indeed the low Rubisco activities reported in this paper are likely due to a combination of fallover and irreversible inhibitor binding, since the RuBP used by the authors is well known to be highly contaminated by inhibitors (described by Ref. 45 in manuscript – Kane et al 1998 as well as Andralojc Biochem J. 2012 442:733-42)

Response 1: We sincerely appreciate the Reviewer’s insightful suggestion, which corrects our misinterpretation of the complete inactivity of RuBisCO after 2 min. After a careful test of the 3-PGA production by free RuBisCO as a function of time when additional fresh RuBP is added

after 5 min, it is found that more 3-PGA is produced as compared to the test without additional RuBP, as shown in Fig. S9. It implies that the stopped production of more 3-PGA after 2 min is indeed not due to the feedback inhibition but the exhaustion of RuBP, as suggested by the Reviewer. The concentration of RuBP used here is 0.5 mM and all the RuBP concentrations have been added into the figure legends.

The production of 3-PGA by free RuBisCO with different reaction times were also determined. The amount of RuBisCO used is 21.875 μg . Production solution I is collected for 21 μL . For the 3-PGA production with the reaction time of 10 min, two tests are conducted to explore the reason of no 3-PGA producing after 2 min-reaction. One test directly uses the production solution I after 10 min for analysis. As shown in Fig. S9, the production of 3-PGA still does not increase. The other one is to add fresh RuBP into the production solution I after 5 min (final RuBP concentration is 0.5 mM) and then to collect the production solution I after another 5 min for analysis. By contrast, more 3-PGA is produced this time, which shows that the stopped increasing of 3-PGA after 2 min is due to the exhaustion of RuBP.

Fig. S9 Production of 3-PGA by free RuBisCO as a function of time. Red circle represents the 10-min 3-PGA production after fresh RuBP is added into production I (final RuBP concentration is 0.5 mM) after 5 min reaction. The amount of RuBisCO used is 21.875 μg . Production solution I is collected for 21 μL . Source data are provided as a Source Data file.

--- Page S10, Section 6 of Supplementary Information

Moreover, in the RIMRs reaction, the products are flowed through the reactor and pumped out immediately after reaction. The accumulation of 3-PGA around RuBisCO is reduced and the control of the reaction time can be achieved straightforwardly by changing the flow rate of the reactant mixture injection (see section 5 of Supporting Information). This is one of the core features of RIMRs.

--- Page 10, paragraph 2, lines 196 – 201

Comment 2: “From the results, similar conclusion can still be obtained. After immobilization, the value of K_m (RuBP) becomes larger and V_{max} becomes smaller. The corresponding k_{cat} of free RuBisCO is about 0.19 s^{-1} , which is indeed smaller than the natural k_{cat} of RuBisCO from Spinach. The major reason may be due to that RuBisCO from Sigma (Catalogue # R8000) is partially purified. The activity given by Sigma is only $0.01\sim 0.1 \text{ unit/mg}$, where one unit can transfer one molecule RuBP and CO_2 to two molecule 3-PGA. In our experiments, the obtained maximum activity would be $\sim 0.08 \text{ unit/mg}$, which is within the range of the reference data given by Sigma”.

You need to explicitly point out in your manuscript that compared to literature the activity of your Rubisco is very low. I suggest using a version of the explanations I have provided above regarding fallover and the documented terrible quality of Sigma RuBP.

Response 2: Following the Reviewer’s comment, more explanations have been added to the manuscript.

It is noted that the activity of free RuBisCO is very low compared with the reported RuBisCO activity before. According to Table 1, the k_{cat} of free RuBisCO here is about 0.19 s^{-1} , which is remarkably smaller than the natural k_{cat} of RuBisCO from Spinach ($\sim 3 \text{ s}^{-1}$)⁴⁸. One reason is likely the irreversible binding of inhibitory contaminants produced during the commercial preparation of RuBP^{49,50}. Another reason for the low activity of RuBisCO here may be the fallover during catalysis, which is caused by the tight binding of inhibitory sugar phosphates to the active site⁵¹⁻⁵³.

--- Page 8, paragraph 2, lines 154 – 160

- 48 Stitt, M., Lunn, J. & Usadel, B. Arabidopsis and primary photosynthetic metabolism—more than the icing on the cake. *The Plant Journal* **61**, 1067-1091, (2010).
- 49 Kane, H. J., Wilkin, J.-M., Portis, A. R. & John Andrews, T. Potent inhibition of ribulose-bisphosphate carboxylase by an oxidized impurity in ribulose-1,5-bisphosphate. *Plant Physiology* **117**, 1059-1069, (1998).
- 50 Andralojc, P. J. et al. 2-Carboxy-D-arabinitol 1-phosphate (CA1P) phosphatase: evidence for a wider role in plant Rubisco regulation. *Biochemical Journal* **442**, 733-742, (2012).
- 51 Edmondson, D. L., Badger, M. R. & Andrews, T. J. A kinetic characterization of slow inactivation of ribulosebisphosphate carboxylase during catalysis. *Plant Physiology* **93**, 1376-1382, (1990).
- 52 Edmondson, D. L., Badger, M. R. & Andrews, T. J. Slow inactivation of ribulosebisphosphate carboxylase during catalysis is not due to decarbamylation of the catalytic site. *Plant Physiology* **93**, 1383-1389, (1990).

- 53 Edmondson, D. L., Badger, M. R. & Andrews, T. J. Slow inactivation of ribulosebiphosphate carboxylase during catalysis is caused by accumulation of a slow, tight-binding inhibitor at the catalytic site. *Plant Physiology* 93, 1390-1397, (1990).

--- Page 26 in References

Comment 3: “Comment 3: Pg. 2 “we also carried out a series of studies on the light-dependent reactions... very high efficiency“ – this is a strange claim in the absence of a reference. Either remove or provide citation. Response 3: The absence of the reference is due to the requirement of double-blind review process. We cannot cite the papers from our group in the manuscript. The citation will be added before the final publication.”

This is a misunderstanding that will need editorial input. It is not acceptable to add references to a manuscript after peer review (since the accuracy of references need to be carefully evaluated by the reviewer), therefore in double blind the authors need to refer to such papers using neutral language (not “we showed”).

Response 3: Thanks for the reviewer’s reminding our misunderstanding. The sentence has been changed and the corresponding references have also been added.

Although the APS has attracted substantial research interest, most of efforts have been devoted to the light-dependent reactions for hydrogen generation and water splitting^{6,7}, microbial growth and biofuel production⁸⁻¹², and photocatalytic cofactor regeneration¹³⁻¹⁶. The exploration of light-independent reactions for food production is worth more attentions.

--- Page 2, paragraph 1, lines 31 – 35

- 13 Lee, J. S., Lee, S. H., Kim, J. H. & Park, C. B. Artificial photosynthesis on a chip: microfluidic cofactor regeneration and photoenzymatic synthesis under visible light. *Lab Chip* **11**, 2309-2311, (2011).
- 14 Liu, J. & Antonietti, M. Bio-inspired NADH regeneration by carbon nitride photocatalysis using diatom templates. *Energy & Environmental Science* **6**, 1486-1493, (2013).
- 15 Huang, X. *et al.* Microfluidic chip-based one-step fabrication of an artificial photosystem I for photocatalytic cofactor regeneration. *RSC Advances* **6**, 101974-101980, (2016).
- 16 Zhao, Y. *et al.* Fully conjugated two-dimensional sp²-carbon covalent organic frameworks as artificial photosystem I with high efficiency. *Angewandte Chemie* **131**, 5430-5435, (2019).

--- Page 22 – 23 in References

Comment 4: Table 1: Explain what are the errors in this Table based on, why are they so small? Please also provide the true source data (time courses) that these numbers are based on in the Excel file, not just processed reaction rates.

Response 4: K_m and V_{max} are calculated by the GraphPad Prism 7 according to the nonlinear fitting of Michaelis-Menten model. Here the errors are standard deviation directly given by the software when the mean reaction rates of three repeated tests under each RuBP concentration are entered as the Y values. If the raw reaction rates (not the mean values) of three repeated tests are entered as the Y values for the calculation, the software would also give the errors, whose values are slightly different from the previous ones. As to the small errors, we have repeated the kinetic experiments at this revision stage and have got similar results and errors as shown in Table 1. The time course data of the new results for Table 1 have been added into the Source Data file.

Table 1: Kinetic parameters obtained from RIMRs reaction and bulk reaction.[&]

	V_{max} (mmol·min ⁻¹ ·g ⁻¹ RuBisCO)	K_m (RuBP) (mM)
RIMRs reaction	0.070 ± 0.003	0.070 ± 0.012
Bulk reaction	0.169 ± 0.006	0.049 ± 0.008

[&] The collected production solutions I and II are both 100 μL. RuBP concentrations are 0.01 mM to 2 mM for the bulk reaction and 0.025 mM to 2 mM for the RIMRs reaction. K_m and V_{max} values are the means ± s.d. of three repeated experiments (n=3), which is calculated by the GraphPad Prism 7 according to the nonlinear fitting of Michaelis-Menten model. Source data are provided as a Source Data file.

--- Page 7, paragraph 1, lines 128 – 133

Comment 5: Line 348: section 4, not session 4

Response 5: Thanks to the reviewer for pointing out. All the “session” mentioned in the manuscript have been corrected to “section”.

The accumulation of 3-PGA around RuBisCO is reduced and the control of the reaction time can be achieved straightforwardly by changing the flow rate of the reactant mixture injection (see section 5 of Supporting Information).

--- Page 10, paragraph 2, lines 198 – 200

The strategy for the PDA modification was adopted from Zheng's group with minor modification⁴⁴ (see section 1 in Supplementary Information).

--- Page 17, paragraph 2, lines 359 – 361

The activities of the free and immobilized RuBisCO were determined using an amplification signal assay adapted from the previously reported method (see section 4 of Supplementary Information)⁵⁶.

--- Page 19, paragraph 1, lines 398 – 400

The production solution I (mixture of RuBisCO, RuBP, HCO_3^- and products) was collected from the outlet of the reactors before it was added into the assay mixture (see section 4 in Supplementary Information) for 3-PGA amount determination.

--- Page 19, paragraph 1, lines 402 – 405

Comment 6: Line 349: Define the substrate concentrations in the reactant mixture. This also applied to Line 102 and 105 of the supplementary information.

Response 6: Sorry for the missing parameter. The RuBP concentration in the reactant mixture is 0.5 mM.

In the immobilized RuBisCO activity assay, reactant mixture (HCO_3^- and 0.5 mM RuBP in the reaction buffer) was passed through the RIMRs at the flow rate of $7 \mu\text{L}\cdot\text{min}^{-1}$ (reaction time is 1 min).

--- Page 19, paragraph 2, lines 400 – 402

For the immobilized RuBisCO activity assay, reactant mixture (HCO_3^- and 0.5 mM RuBP in the reaction buffer) was passed through the RIMRs and the production solution I (containing RuBisCO, RuBP, HCO_3^- and 3-PGA in the reaction buffer) was collected from the outlet of the reactors for further assay.

--- Page S8, lines 99 – 102 of Supplementary Information

REVIEWERS' COMMENTS:

Reviewer #3 (Remarks to the Author):

The changes made address my comments satisfactorily. As a final minor change please also state the bicarbonate concentration used at every instance where RuBP concentration was inserted. (Substrate concentration refers to both RuBP and HCO₃⁻)

Response to Reviewer 3

General comment: The changes made address my comments satisfactorily. As a final minor change please also state the bicarbonate concentration used at every instance where RuBP concentration was inserted. (Substrate concentration refers to both RuBP and HCO₃⁻).

Response: We sincerely appreciate the Reviewer's satisfactory on our modification. In addition, following his suggestion, we have added the bicarbonate concentration at every instance where RuBP concentration was inserted in the manuscript and Supplementary Information.